# Human-robot collaborative task planning using anticipatory brain responses

**Stefan K. Ehrlich**[1]*, **Emmanuel Dean-Leon**[2], **Nicholas Tacca**[3], **Simon Armleder**[1], **Viktorija Dimova-Edeleva**[4], **Gordon Cheng**[1,5]

1 Chair for Cognitive Systems, Department of Electrical Engineering, TUM School of Computation, Information and Technology, Technical University of Munich, Munich, Germany, 2 Department of Electrical Engineering, Automation, Chalmers University of Technology, Göteborg, Sweden, 3 Battelle Memorial Institute, Columbus, OH, United States of America, 4 MIRMI - Munich Institute of Robotics and Machine Intelligence, formerly MSRM, Technical University of Munich, Munich, Germany, 5 Center of Competence NeuroEngineering, Technical University of Munich, München, Germany

* stefan.kh.ehrlich@gmail.com

**Data Availability Statement:** The data is available in the repository https://github.com/stefan-ehrlich/HRC_neurobased_taskplanning.git.

## Abstract

Human-robot interaction (HRI) describes scenarios in which both human and robot work as partners, sharing the same environment or complementing each other on a joint task. HRI is characterized by the need for high adaptability and flexibility of robotic systems toward their human interaction partners. One of the major challenges in HRI is task planning with dynamic subtask assignment, which is particularly challenging when subtask choices of the human are not readily accessible by the robot. In the present work, we explore the feasibility of using electroencephalogram (EEG) based neuro-cognitive measures for online robot learning of dynamic subtask assignment. To this end, we demonstrate in an experimental human subject study, featuring a joint HRI task with a UR10 robotic manipulator, the presence of EEG measures indicative of a human partner anticipating a takeover situation from human to robot or vice-versa. The present work further proposes a reinforcement learning based algorithm employing these measures as a neuronal feedback signal from the human to the robot for dynamic learning of subtask-assignment. The efficacy of this algorithm is validated in a simulation-based study. The simulation results reveal that even with relatively low decoding accuracies, successful robot learning of subtask-assignment is feasible, with around 80% choice accuracy among four subtasks within 17 minutes of collaboration. The simulation results further reveal that scalability to more subtasks is feasible and mainly accompanied with longer robot learning times. These findings demonstrate the usability of EEG-based neuro-cognitive measures to mediate the complex and largely unsolved problem of human-robot collaborative task planning.

## 1 Introduction

Scenarios in which assistive robots are needed are more common and present in multiple domains. These domains include service robots in hospitals [1–3], households [4], and Cobots [5] in flexible industrial manufacturing. Human-robot interaction (HRI) is essential for the future applications of robots in these domains due to the increasing demand for adaptability

**Funding:** The author(s) received no specific funding for this work.

and flexibility. HRI can be defined as a joint activity between humans and robots in a shared symbiotic working environment, with the primary objective to accomplish *together* a set of tasks [6]. HRI is a process in which an autonomous robot system and a human operator work on simultaneous tasks within a collaborative environment. Examples of such collaborations are scenarios where the robot and human perform complementary tasks, either because of the human partner's preferences or because both partners have complementary strengths and weaknesses. For example, assembly operations where the robot manipulates heavy objects while the human partner performs tasks requiring dexterous manipulations. In these scenarios, constant task switching and takeovers are required between the robot and the human, which must be solved smoothly and safely. In particular, for industrial scenarios, the aim is to increase functionality and improve performances. Overall, HRI aims to generate robot *transparency* in which the deployment of the robot does not produce disruptions to the environment. Ideally, a robot can be considered fully transparent when the deployment of the robot does not produce disruptions (changes) in the environment. Robot Transparency can be measured by the effort needed to deploy the robot, such as safety mechanisms, personnel training, and changes in the process. The more changes needed, the less transparent the robot is [7]. Thus, the robot should be both aware of its environment and facilitate the user awareness to be transparently integrated, e.g. it must understand the actions, preferences, and intentions of its co-workers, and be able to inform its intentions, reasoning, future plans, and the potential uncertainties [8].

## 1.1 Open challenges in HRI

Genuine HRI with the efficiency and fluency [9] of human-human collaborations is a hard and complex problem. In fact, the relationship between humans and robots is not yet fully understood and there are still confusions on the role assignment and task ownership [6]. For example, when should a robot take the leader or the follower role? What should be the determining factor in assigning tasks to a robot or a human, in particular, when both agents can perform the same task? What should be the robot's autonomy level? When should the user be *in-the-loop* and *out-of-the-loop* [10]? All of these criteria can change under different conditions, e.g. Parasuraman et al. [11] provides a study of adaptive vs static task allocation in the aviation.

One of the essential research fields to solve this problem is task planning with role assignment for joint human-robot tasks. This task planning defines the timing and order of the robot and human actions, i.e., when the robot should provide assistance to the user. In deterministic scenarios, this planning can be pre-designed. However, in more natural interaction scenarios, task planning should be flexible to accommodate human preferences and online decisions. This flexibility comes naturally in human-human interactions, especially when the actors share the same goal, since humans are equipped with the ability to process the required information to infer and accommodate each others' preferences [12]. However, achieving the same level of fluency and comfort in HRI presents many complex challenges. For example, the robot needs to adapt the way it assists the human collaborator, i.e., adapt its behavior. Therefore, the robot must adopt a situational and context-based approach to adapt its behavior. The assistance activation is usually shaped by a pre-defined policy. The most usual policies are: to minimize human-idle time [13]; to reduce human cognitive and physical load [14]; to produce a low-level motion planning within a collaborative context, including human-aware motions with early predictions to estimate the best robot's actions [15]; to improve team fluency and user's sense of safety [15, 16]; to coordinate actions during execution time based on low-level behaviors, e.g. gaze cues to shape collaboration roles [17], or to coordinate interactions, such

as handovers [18]; methods using verbal-feedback from the user [19], or sharing common resources during collaborative tasks [20]; hybrid policies combining human-initiated (explicit request), robot initiated reactive (when help is needed), and robot-initiated proactive collaborations (help whenever possible, non-conflicting simultaneous/parallel tasks) [21, 22].

The prevalent foundation for these different policies is a *complex* artificial cognitive system capable of detecting, estimating, inferring and learning human actions, intentions, and preferences [23–28]. Results from the literature strongly suggest that anticipatory perceptual information improves fluency in collaborative work [29]. In addition to this complex artificial cognitive system, there are important aspects to guarantee efficiency in HRI, e.g. eliciting trust [10, 30–32] and acceptance in the human co-worker [33] through situation awareness [8]. Researchers have devoted much effort to develop systems that increase safety during HRI, e.g. vision-based systems [34]. Furthermore, the user must be provided with intuitive interfaces that help to communicate with the robotic system, achieving physical and cognitive interactions [8, 35]. Usually, these interfaces involve high-level multi-modal commands, including natural language processing [36], gesture and posture tracking [37], and haptic interactions [38]. All of these technological and scientific challenges/requirements render an extremely complex problem to solve. In terms of HRI, two critical situations pose important open questions:

- The human chooses or claims responsibility for a subtask without the robot being aware of or explicitly informed about. How can a robot detect this task selection done by the user?

- The human assumes the robot to be responsible or needs its assistance in taking over a subtask without explicitly asking for help. How can a robot detect this task assignment without being explicitly requested by the user?

Regarding the task planning problem, the first situation is related to tasks that a user expects or is planning to take. Smooth task planning requires the robot to infer the human's internal/personal choices on the spot; an issue which is currently largely unsolved in HRI. The second question is related to tasks that a user prefers to be taken by the robot. Smooth task planning requires the robot to assess the level of agreement from the user on the spot when assisting with a subtask; an issue which is likewise largely unsolved in HRI. Addressing these challenges provides one step forward to genuine HRI. However, the current state-of-the-art in robot sensory and reasoning technology makes it difficult, in some cases even impossible, to swiftly infer humans' internal states and beliefs, such as reliably detecting humans' preferences and intentions based on visual information alone.

## 1.2 Neuroengineering approaches to HRI

Novel avenues of research and applications at the intersection of robotics and neuroscience are recently gaining traction and progressively starting to make impacts on modern robotics [39, 40]. Brain-based systems that follow the neuroergonomics approach are more promising than systems that use models connecting behavior and workload [41]. In particular, human-in-the-loop approaches using implicit human feedback have shown to reduce the complexity of the robotic system and mediate the need for more advanced robot perception and reasoning capabilities. The basic idea is to derive information about the human partner's bodily or cognitive state in near real-time from electro- or psychophysiological signals such as electroencephalography, electromyography, eye-tracking, heart-rate, or galvanic skin response, or a combination thereof (see for instance [42]) without the user being necessarily aware of it. This information can then be provided to the robot during the interaction for updates to reflect the human's preferences. An example of this is a passive brain-computer interface (BCI) [43],

which enables the augmentation of the information accessible through contemporary robot perception and reasoning technology by an additional channel that taps directly into the human partner's cognition. Besides the assessment of the user's mental state, this approach has been shown to be useful for the assessment of the user's subjective perception and evaluation of the robot's behavior during HRI, as well as the online adaptation of the robotic system to facilitate or even improve the interaction/collaboration with the human partner [44] (see also [45] for a recent review). More specifically, we showed in a pilot study, the feasibility to decode subjects' intentions to engage in eye contact with a humanoid robot from the ongoing EEG of the human partner while interacting with the robot. This study also showed the feasibility to decode the human partner's believed role during the interaction, e.g. whether the human believed to have been the initiator or follower of an eye-contact event [46]. The use of neuro-cognitive measures as an implicit feedback signal for robot adaptation to facilitate HRI has been shown also by Szafir and Mutlu [47] in which a humanoid robot appeared as a story narrator. Based on EEG measures of attention / task engagement of the human observer, the robot adapted its level of gesticulation, mimics, and gazing during storytelling, which in turn positively influenced the level of details the human subject could recall of the story after the experiment.

Other works have focused on anticipation- and error-related EEG signals. While the former is a motor preparatory signal prior to movement onset which is manifested as a slow negative wave over fronto-central areas in anticipation of an event, the latter is a phenomenon related to error- and performance monitoring [48], observable upon human observation of an erroneous or unexpected event and manifested in a specific event-related potential (ERP) following the observed event [49–51]. Both phenomena have been studied in the context of BCI and HRI. Anticipatory movement related potentials have been shown to be decodeable in single trial in anticipation of upcoming events and demonstrated to be useful to improve BCIs or HRI [52–56]. The error-related potential (ErrP) is evaluative in its nature, e.g. arises as a response to an observed action, independent of whether this action was self-inflicted or executed by an external instance, such as a robot. ErrPs have been proposed and successfully validated as a useful measure for the implicit assessment of erroneous or unexpected robot actions from the viewpoint of the human partner [57–59] and demonstrated as an implicit human feedback signal for robot adaptation during HRI. ErrP-based robot adaptation was shown in several case studies, such as in the context of robot learning of a human desired end-effector trajectory by Iturrate and colleagues [60], a binary sorting task by Salazar-Gomez and colleagues [57], the recognition and replication of human gestures by Kim and colleagues [61], mediation of human-robot co-adaptation [62], the inference of subject preferences on robot trajectories by Iwane and colleagues [63], and very recently an example of robot learning control-policies for singularity avoidance by Aldini and colleagues [64].

## 1.3 Aim of this work

In the present work we explore the feasibility of using EEG-based neuro-cognitive measures for online robot learning of dynamic subtask assignment. We focus here on a dyadic interaction between human and robot where both partners have equal or similar levels of responsibility for the joint task at hand. To this end, the present work employs a *real physical robotic system* (UR10 industrial robotic manipulator) for the featured HRI to be as close as possible to realistic joint human-robot tasks. The joint tasks featured in this study allude to industrial co-working scenarios, e.g. human-robot and robot-human handover situations in assembly tasks; mixed or shared responsibilities in which partners are responsible for different areas of the workspace or different subtasks. In summary, this research provides two main contributions:

- This work demonstrates in an experimental human subject study the presence of EEG measures indicative of a human interaction partner anticipating takeover situations from the human to the robot or vice-versa. These measures are further shown to be decodeable from the human EEG in single-trials.

- This work further proposes a reinforcement learning based algorithm employing these anticipatory measures as a neuronal feedback signal to the robot for dynamic learning of subtask-assignments, and finally demonstrates its feasibility in a simulation-based study.

This paper is structured as follows: Section 2 describes the methodology, data analysis, and results of the experimental HRI study. Section 3 describes the methodology, data analysis, and results of the simulation-based study. Section 4 jointly discusses the result and findings of both studies, and Section 5 concludes the work.

## 2 Experimental HRI study

### 2.1 Participants

Twelve healthy subjects participated in the experiment. The data of one subject (s07) had to be excluded from further analyses due to heavy contamination of the EEG signals. The remaining eleven subjects were approximately gender-balanced (5 female, 6 male) with 26.8±3.7 years of age (M±SD; M = mean; SD = single standard deviation). Since the focus of the work lies on single-subject effects, a larger sample size is not required. Nevertheless, the sample size was chosen >10 to allow also for basic statistical analyses and tentative conclusions on group level. All subjects had a technical educational background with the majority at bachelors degree level, e.g. electrical engineering, computer science, or mechatronics. All but one subject (s05) were right-handed and all subjects had normal or corrected to normal vision. The participants were equally instructed about the experiment protocol, provided written informed consent regarding participation in the experiment, and were compensated for their efforts after the experiment. The study was approved by the institutional ethics review board of the Technical University of Munich under reference number 769/20 S-KK.

### 2.2 Experimental paradigm

The experimental task comprised the interaction between a human and a robot in completing a trajectory-following task in a 7x7 grid-world environment (Fig 1) which is an adaptation from our previous work [65]. To successfully complete the trajectory, human and robot were tasked with moving the robot end-effector to each grid tile within the trajectory in sequential order. A movement of the end-effector from one grid tile to the next grid tile is considered a *trial*. The start and end locations of the target trajectory were selected randomly on the inner 5x5 grid such that at least 7 trials (on average 7.19±0.39) were required to complete the trajectory. There was no requirement for the trajectory to follow the shortest path. The tiles on the border were excluded from the highlighted trajectory to allow for movement in the four possible directions. The trajectory between the start and end positions was connected with the tiles in between including at least one direction change. A single executed trajectory was termed *episode*. In the beginning of an episode, the robot automatically moved to the start position and shortly thereafter the trajectory appeared on display. Throughout the experiment, human and robot have clearly defined roles and responsibilities for subtasks. The experiment featured two interaction scenarios. The first scenario is the *sequential Collaboration (sC)* scenario (Fig 1A), in which human and robot are each responsible for a certain area in the workspace. Each area contained 19–30 grouped tiles with one group colored in green and the other in blue, one being assigned to the human and one to the robot. The assignment switched from time to time

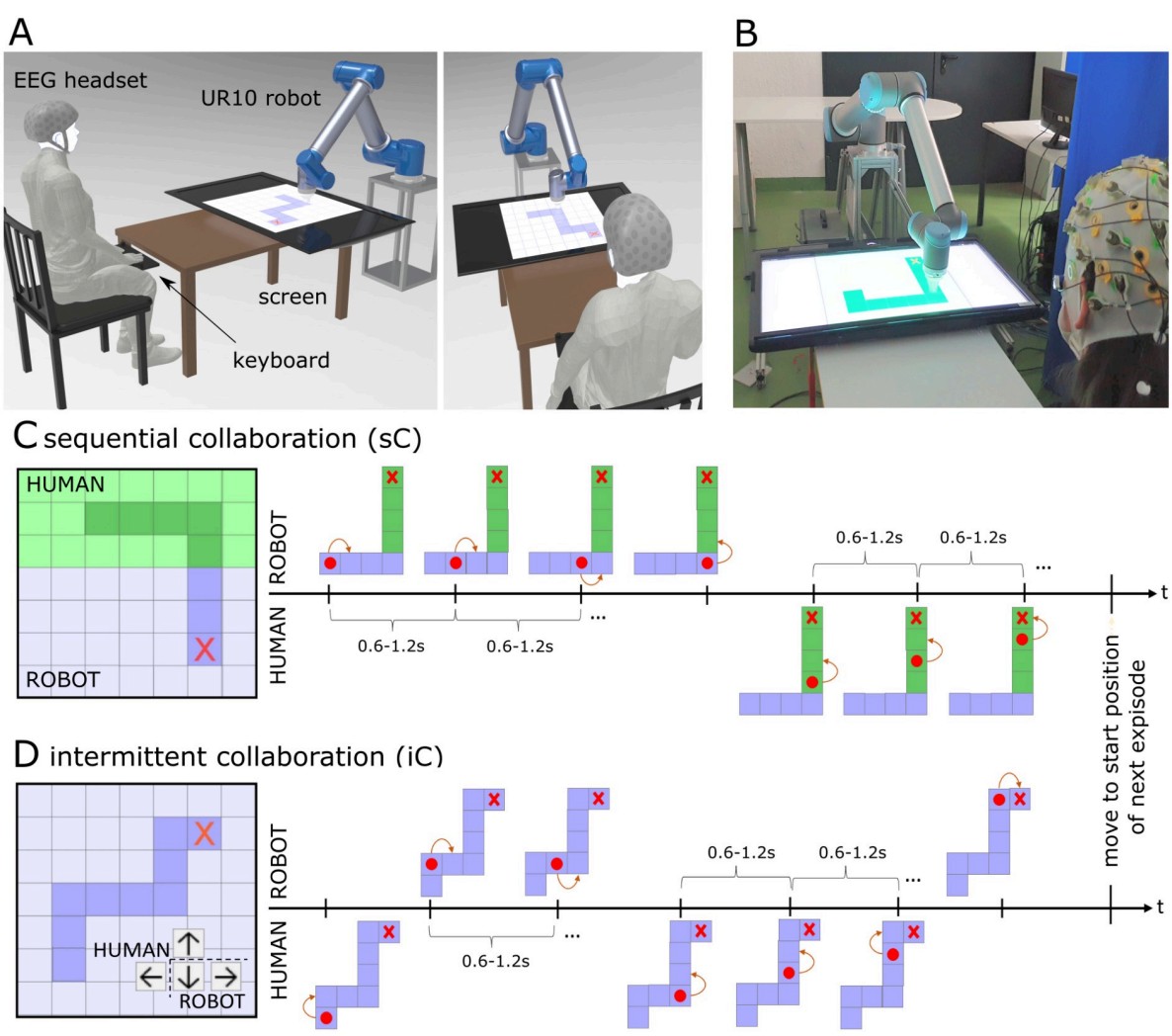

**Fig 1. Experimental paradigm and setup.** Experimental paradigm featuring a trajectory-following task with two interaction scenarios. Panel A and B: Experimental setup showing a subject sitting in front of the industrial robot UR10. The workspace is displayed on a computer screen located between the human and the robot. The robot end-effector determines the current location in the trajectory. The subject indicates commands to the robot via key-presses. Panel C: *sequential Collaboration (sC)* scenario, in which human and robot are each responsible for a certain area in the workspace (left), and panel D: *intermittent Collaboration (iC)* scenario in which both human and robot are responsible in the entire space, but each having control over two out of four directions (right).

to avoid habituation and color sensitivity. This scenario simulates a sequential collaborative takeover situation, in which the responsibility changes at the border of the colored areas. The second scenario is the *intermittent Collaboration (iC)* (Fig 1B) in which both human and robot are responsible for the entire trajectory. However, both the robot and human are limited to controlling two of the four directions. Together they are able to control the robot end-effector in all four directions when combining each of their respective movement options. In the particular example shown in Fig 1B, the human controls the *up* and *left* directions, while the robot is responsible for the *down* and *right* directions. This task simulates scenarios when the human and robot are responsible for complementary subtasks, for instance, if the human partner has specific preferences, or both partners have complementary strength/weaknesses. Takeover situations occur intermittently depending on the available movements, robot end-effector

position and the next grid position within the target trajectory. In both scenarios, takeovers occurred one after the other with no limit for the maximum number of takeovers, i.e., there could have been several takeovers over the border in the *sC* scenario and several direction changes, both for *sC* and *iC*. The human could control the movements only in the designated area in the *sC* scenario and only along the two possible directions in the *iC* scenario. The robot could control the complementary cases, and beyond them, the robot could mistakenly take-over in situations when it was not necessary for a takeover. Robot mistakes occurred at a rate of 30% across the whole workspace. Not all mistakes resulted in a takeover from the human or even forced a subsequent takeover to occur. The resulting correction could still be the responsibility of the robot, but it could also be the responsibility of the human depending on the direction the robot moved with respect to the next grid position in the trajectory. The explicit definition of tasks and the time/place where they will be triggered fosters the mental preparation of the user and induces expectations and thus *anticipations* of upcoming takeover situations. Trials in which these anticipations have likely occurred, e.g. shortly before a takeover situation, are of central interest in the subsequent data analysis.

## 2.3 Stimuli and apparatus

**2.3.1 Experimental setup.** Subjects were seated directly across from a 6 degree of freedom industrial robotic arm (UR10-CB3, Universal Robots) with an LCD 42" monitor face up on a small table positioned between the subject and robot. The end of the table was positioned 0.8 m from the robot base in such a way that the two edges of the grid on the screen were 0.76 m and 1.25 m respectively from the robot base. The table was 1 m long with the subject positioned directly at the end of the table approximately 1.8 m away from the robot base seated facing forward with the head approximately parallel to the ground at a visual angle to the monitor between 30 degrees to 35 degrees (Fig 1A and 1B). This distance ensured that subjects were far enough away from the robot so that at its full extension, the robot could not physically harm the subjects. Additionally, the monitor and table combined height was 0.6 m from the ground with the display and robot actions easily visible. The end-effector of the robot was positioned at a constant height of 2 cm above the monitor and moved in the $X − Y$ plane of the screen to the different grid locations. At no point did the robot move outside of the experiment grid. As a result, robot movements both occurred at a safe distance away from and below the subjects direct eye sight level to reduce intimidation from robot movements toward the subject.

The experiment grid was displayed in full screen so that it had a size of 49 cm x 49 cm with each grid tile having a size of 7 cm x 7 cm. Depending on the current situation, subjects either observed the robot motions passively or controlled the robot via directional arrows on a keyboard. All other unnecessary keys were removed from the keyboard to keep the interface as simple as possible. Subjects were instructed to rest the keyboard on their lap and to maintain a relaxed position throughout the experiment. Discrete key presses commanded the robot to move to the neighboring grid position in the corresponding direction. The robot was programmed to not move to the next position until it was within a 2 mm radius of the goal end-effector position at the center of the grid space. If subjects pressed a key before the robot was within the threshold, the robot would not process the command and subjects were instructed to press the key again once the robot reached position. The average time duration of one trial for the *sC* scenario was 875.88±133.91 *ms* (M±SD) and 872.34±131.35 *ms* for the *iC* scenario.

**2.3.2 Robot control.** The motion of the robot's end-effector within the $X − Y$ plane was constrained at a fixed height above the screen. The orientation was kept constant and pointed toward the screen. To smoothly move the end-effector from one tile to another, we planned trajectories $\mathbf{x}(t)$ that minimize jerk, i.e., time-derivative of acceleration. The resulting

trajectories are desirable for their similarity to human joint movements and to limit robot vibrations [66]. Choosing $\mathbf{x}(t)$ as a 5th-order polynomial function ensures a minimum jerk [66]. By adding zero velocity and acceleration constraints at the beginning and end of the motion we get the following:

$$\mathbf{x}(t) = \mathbf{x}_i + s(t)(\mathbf{x}_f - \mathbf{x}_i) \tag{1}$$

$$s(t) = \frac{6}{t_f^5} t^5 - \frac{15}{t_f^4} t^4 + \frac{10}{t_f^3} t^3 \tag{2}$$

where $t_f$, is the total duration of the movement trajectory, and $\mathbf{x}_i$, $\mathbf{x}_f$ are the initial and final position of the robot's end-effector, respectively. In our experiments, we set the robot movement duration at the beginning of a session to a time point randomly chosen within a $t_f \in [0.6 \text{ s} - 1.2 \text{ s}]$ time range. A combination of preceding events and the 2mm end position radius threshold induced random deviations from the robot movement duration per each trial, which helped account for subject habituation to robot movement onset. Finally, the desired trajectory was fed into a Cartesian tracking controller that determines the required joint velocities of the robot [67]. Commands were sent to the real-time interface of the UR10 robot at a fixed control loop frequency of 125 Hz. The control system was implemented with the Robot Operating System (ROS Noetic) [68] and executed on a dedicated AMD Ryzen 7 3800x computer, running Ubuntu 20.04.

**2.3.3 EEG data acquisition and pre-processing.** EEG data was acquired with a Brain Products actiChamp amplifier equipped with 32 active EEG gel-based electrodes arranged according to an extended international 10–20 system [69] (FP1, FP2, F3, F4, F7, F8, FC1, FC2, FC5, FC6, C3, C4, T7, T8, CP5, CP6, P3, P4, P7, P8, TP9, TP10, O1, O2, Fz, Cz, Pz, EOG1, EOG2, EOG3). All leads were referenced to the average of TP9 and TP10 (average mastoids referencing) and the sampling rate was set to 1000 Hz. Three channels were used for capturing electrooculogram (EOG1–3) signals in three locations of the participants face (left and right outer canti and forehead) according to a method suggested by Schloegl and colleagues [70]. The EEG amplifier was battery-driven and located on a table nearby the participant. The data was transferred via USB to a separate recording PC (Intel Core TM i5 CPU 750@2.67 GHz). The amplifier was connected to the PC executing the experiment protocol via parallel port over which event triggers were transmitted to be stored synchronously with the EEG signals.

The subsequent EEG data pre-processing steps were applied for data cleaning and conditioning in part using functions provided by the Matlab EEGLAB toolbox [71]: In order to remove high frequency and power-line noise, we first filtered the signals of the EEG and EOG channels using a zero-phase Hamming-windowed sinc FIR band-pass filter with cutoff frequencies of 1 Hz and 40 Hz. EOG activity in the EEG signals (eye-blink and lateral eye movements) was corrected using a regression-based method by Schloegl and colleagues [70]. Next, we identified contaminated EEG channels using normalized kurtosis with a threshold of 2 (std. dev.) and subsequently used spherical interpolation to reconstruct rejected channels from the signals of neighboring electrodes. Following this procedure, for each subject between 3 to 6 channels were replaced. Afterward, the data were re-referenced to a common average reference (CAR) to minimize signal contamination from external noise sources. Finally, the data was down-sampled to 512 Hz to reduce computation time in all further processing steps. Please note that data processing was performed offline, but each step of the processing pipeline can be executed online (e.g. in near real-time) as well, if the band-pass filter is replaced with a causal filter. See example in one of our earlier works featuring a closed-loop BCI study [62].

**2.3.4 Timing validation.** In light of the planned EEG data analyses, the precise monitoring of the timing of the onset of robot motion was critical in the experiment. To exclude the presence of any harmful system latency or timing jitters between robot motion onset and parallel port event trigger, a single validation experiment was conducted prior to any subject participation. The results revealed timing latencies and jitters small enough to not compromise the planned EEG data analyses. The exact methodology of the timing validation experiment is described in S1 Appendix.

## 2.4 Experimental protocol

The experiment consisted of 12 blocks in total, alternating between *sC* and *iC* scenarios with a total of 13 episodes per block. An episode contained all trials, i.e. single discrete movements from one to an immediately neighbouring tile, required to arrive from the start to the end, following the marked trajectory on the grid. Following a successful completion of an episode, a new episode started with new trajectory. After the EEG setup was prepared, subjects first conducted a single test block of each scenario consisting of five episodes per scenario to ensure the full understanding of the experiment paradigm. Next, the experiment began with subjects working collaboratively with the robot and taking short breaks between blocks as desired. Questionnaires were given before and after the experiment to account for possible outliers in the data based on personality traits and their familiarization with robots. Scores from the questionnaires did not reveal any confounding information and were thus not analyzed further.

## 2.5 Data analysis of human-robot takeover situations

The experiment features four distinct situations in which takeover of responsibility, either from robot to human or from human to robot, takes place. In the sequential collaboration (*sC*) scenario, these situations occur at the border of the workspace division (see Fig 1C), whereas in the intermittent collaboration (*iC*) scenario, these situations may happen throughout the entire trajectory whenever the previous action was performed by the robot (*R*) and the next by the human (*H*) or vice-versa (see Fig 1D). These situations can be formulated as four distinct cases of takeover (*HR*, *RH*) or non-takeover (*HH*, *RR*) situations as depicted in Table 1.

**2.5.1 EEG data segmentation.** The data of all subjects were segmented into epochs by extracting time intervals of -200 to 1200 ms relative to the onset of a robot movement (onset of robot movement denotes $t = 0$ms). Out of the extracted epochs, exclusively non-error trials were used for subsequent analyses to avoid mixing anticipatory with error-related brain responses. Epochs were further separated into eight categories according to the defined conditions shown in Table 1. Accordingly, per subject we extracted the following average number of trials per condition: 611±43 (M±SD) trials for each $HH_{tr-1}$ and $HH_{tr}$; 148±14 trials for each $HR_{tr-1}$ and $HR_{tr}$; 530±25 trials for each $RR_{tr-1}$ and $RR_{tr}$, and 212±22 trials for each $RH_{tr-1}$ and $RH_{tr}$. The reason for the imbalanced number of trials per condition is that conditions occurred randomly depending on the current situation during collaboration (shape and size of

**Table 1. Overview of human-robot takeover situations.** Variants of takeover (*HR*, *RH*) and non-takeover situations (*HH*, *RR*) focusing on the action before ($tr - 1$) and after ($tr$).

| case | trial $tr$−1 (before) | trial $tr$ (after) | description |
|------|----------------------|--------------------|-------------|
| *HH* | human ($HH_{tr-1}$) | human ($HH_{tr}$) | human continues |
| *HR* | human ($HR_{tr-1}$) | robot ($HR_{tr}$) | robot takeover from human |
| *RR* | robot ($RR_{tr-1}$) | robot ($RR_{tr}$) | robot continues |
| *RH* | robot ($RH_{tr-1}$) | human ($RH_{tr}$) | human takeover from robot |

trajectory, random selection of areas and subtasks, etc.). The methods in all subsequent analyses were chosen to be robust against this imbalance when contrasting the conditions. The conditions were further separated into the two collaboration tasks (*sC* or *iC*), with approximately half of the trials belonging to the *sC* and the other half to the *iC* scenario. Afterwards, all epochs were investigated for implausibly large amplitudes $> 1000V$, caused by artifacts which could not be filtered during the data pre-processing. This processing step revealed one outlier subject (s07) with many epochs containing such implausibly large signal amplitudes. This subject was subsequently excluded from all further analyses. The data of all other subjects were below $1000V$, such that no epochs had to be excluded. The data analyses comprise three complementary steps: a spatio-temporal analysis of event-related potentials (ERP) (Section 2.5.2) according to the methodology proposed by Luck [72], a time-frequency analysis using the event-related spectral perturbation (ERSP) technique (Section 2.5.3) according to [73, 74] which complements the ERP analysis in revealing the corresponding frequency components and allowing the distinction of time- and phase-locked components. The final analysis investigates the single-trial decoding performance (Section 2.5.4) according to the methodology used in our previous works, e.g. [59].

**2.5.2 Spatio-temporal analysis of event-related potentials (ERPs).**   We investigated the spatio-temporal activation pattern of the ERPs arising while the subject was observing the robot during the various conditions defined in the previous sections. Analyses of the shape and timing of the potentials in each condition was carried out through the computation of the baseline-corrected time-locked grand average potential per condition following the standard procedure [72, 74]. Baseline correction was performed for each trial and channel separately by subtracting the average amplitude of the period -200 to 0 ms from the entire signal epoch. Statistical testing for condition differences was performed using non-parametric permutation tests with correction for multiple comparisons (see methodological details in S2 Appendix).

**2.5.3 Analysis of event-related spectral perturbation (ERSP).**   The ERP analysis is limited because it is only sensitive to effects which are both time- and phase-locked. In addition, it depicts a conglomerate of the spectral components which emphasizes the low frequency components (due to the typical $1/f$ frequency characteristic of EEG signals) without allowing a differentiated view on the underlying spectral components. Therefore, the ERP analysis was complemented with a time-frequency analysis, according to the event-related spectral perturbation (ERSP) technique [73, 74]. To obtain the grand average ERSP per condition, we computed the complex Morlet wavelet transform (CWT) per each trial, channel, and condition. The Morlet wavelet was chosen as it is the most established type of wavelet for EEG signals and thus recommended [74]. The wavelet kernel length was set to 1 second. The frequency range was linearly spaced and set to 2–30 Hz with a resolution of 2 Hz to cover the most relevant frequencies. The number of wavelet cycles was set to gradually increase from 2 cycles for the lowest frequency to 10 cycles for highest frequency. A reasonable trade-off between time- and frequency-resolution was achieved with these parameter choices (kernel length and evolution of number of cycles). From the complex ERSP, the magnitude was computed by taking the absolute value of the complex number and multiplying it by a factor of 2. Finally, the time domain was down-sampled by a factor of 2 to reduce computational load in all further processing and analysis steps. The down-sampling has no effect on the time resolution because it was performed after CWT computation. ERSPs of individual trials were finally averaged without baseline correction. Statistical testing evaluating between condition differences was performed using non-parametric permutation testing with correction for multiple comparisons (see methodological details in S2 Appendix).

**2.5.4 Single-trial decoding of anticipated takeover situations from ERPs.**   The employed classification approach is largely identical to the method employed in our earlier works [59, 62,

75] as it has repeatedly shown to achieve high decoding accuracy for the single-trial classification of ERPs.

*Feature extraction*: In the single-trial classification of ERPs, our previous work and others have shown time-domain features to outperform other types of features [75–77]. Therefore, time-domain features were used in this work. For each trial and each channel, the time-series was first normalized by subtracting the mean within the period 0–800 ms in order to remove any unwanted DC fluctuations. Next, the amplitude of the time-series was averaged within four 50 ms-long windows within the period 400–600 ms and eleven overlapping 100 ms-long windows within the period 0–600 ms, relative to the onset of a robot movement. Since inter-trial times varied from 600–1200 ms, no time-points after 600 ms are considered to ensure that no information from the next trial is included. The shorter windows were chosen to cover the significant component at 500 ms found in the previous analysis in a fine-grained manner; the longer windows were chosen to cover the whole ERP time course in a coarse manner. The resulting feature vector had a length of 405 (27 channels x 15 windows).

*Binary classification*: The classifier used in the analysis was a regularized version of the linear discriminant analysis (rLDA) [78]. The rLDA classifier has been established as a robust method to discriminate mental states based on EEG signals in the field of BCI [79]. The LDA discriminant function is the hyperplane discriminating the feature space corresponding to two classes: $y(\boldsymbol{x}) = \text{sign}(\boldsymbol{w}^{\text{T}}\boldsymbol{x} + b)$, with $\boldsymbol{x}$ being the feature vector, $\boldsymbol{w}$ being the normal vector to the hyperplane (or weight vector), $b$ the corresponding bias, and $y(\boldsymbol{x}) \in \{-1, 1\}$ the classifier decision. The weight vector and bias were computed by $\boldsymbol{w} = (\hat{\boldsymbol{\mu}}_1 - \hat{\boldsymbol{\mu}}_2)(\tilde{\boldsymbol{\Sigma}}_1 + \tilde{\boldsymbol{\Sigma}}_2)^{-1}$ and $b = -\boldsymbol{w}^{\text{T}}(\hat{\boldsymbol{\mu}}_1 + \hat{\boldsymbol{\mu}}_2)$, with $\hat{\boldsymbol{\mu}}_j$ being the class-wise sample means, and $\tilde{\boldsymbol{\Sigma}}_j$ being the class-wise regularized covariance matrices. The indices 1 and 2 represent class 1 and 2, respectively, and are placeholders for the specific classes described in more detail in the next paragraph. The classifier decision $y = -1$ indicates the decision for class 1 whereas $y = +1$ indicates the decision for class 2. Regularization aims at minimizing the covariance estimation error by penalizing very small and large eigenvalues. This leads to robust covariance estimates even for high dimensional feature spaces [79] as in our case. The regularized covariance matrices were computed by $\tilde{\boldsymbol{\Sigma}}_j = (1 - \lambda)\boldsymbol{\Sigma}_j + \lambda\boldsymbol{I}$, with $\lambda \in [0, 1]$ being the regularization parameter, and $\boldsymbol{I}$ the identity matrix. Our previous works [59, 62] featuring similar classification problems have shown that choosing $\lambda$ values close to 1 lead to the best and most robust (unbiased) classification performance. Therefore, the regularization parameter was not optimized, but fixedly chosen at $\lambda = 0.9$ across all subjects and classification tasks.

*Validation*: The above described modeling approach was applied to two separate binary classification problems: the discrimination of an anticipated human takeover from robot versus human continuation ($HR_{tr-1}$ vs. $HH_{tr-1}$) and the discrimination of an anticipated robot takeover from human versus robot continuation ($RH_{tr-1}$ vs. $RR_{tr-1}$). Both classification problems were further separately validated within the two collaboration scenarios using a within subject 10-times-10-fold cross-validation scheme. Per subject, the trials were randomly split into 10 folds with 9 folds used for model calibration and the remaining fold for testing. This procedure was repeated until all folds were used once for testing. The entire procedure was furthermore repeated 10 times. Each time and fold, the number of trials per class of the calibration data was balanced by random pick and replace (please note that the number of trials per class was initially imbalanced). This analysis provides an estimate of how well a participant-specific decoder would perform in classifying unseen data when being calibrated with different data within the same session. Individual classification results per time and fold were averaged and reported per subject as percentage of correctly classified instances (trials/events).

Decoding performance is reported as percentage of correctly classified instances/trials per class as $TNR_{HH}$ and $TPR_{HR}$, as well as across classes as $ACC_{HHvsHR}$ for the discrimination of an anticipated robot takeover from human versus human continuation. Analogously, decoding performance is reported as correctly classified instances/trials per class as $TNR_{RR}$ and $TPR_{RH}$, as well as across classes as $ACC_{RRvsRH}$ for the discrimination of an anticipated human takeover from robot versus robot continuation. In addition, the results of both binary classification problems are reported as overall accuracy.

## 2.6 Results of experimental HRI study

**2.6.1 Electrophysiological responses in human-robot takeover situations.** The results of the analysis of electrophysiological responses to anticipated takeover situations are depicted in Figs 2 and 3. Fig 2 shows the grand average ERP time courses over channel Cz time-locked to the onset of robot movement for each of the four cases as defined in Table 1. In each case, we investigated the moment before (Fig 2A and 2C) and after (Fig 2B and 2D) an anticipated takeover situation. In the analysis of ERPs, significant differences between conditions were only found in contrasting an anticipated human takeover from robot versus robot continuation (Fig 2C). Significant differences were found over prefrontal channels around 500ms after the onset of the robot movement before the anticipated takeover. The amplitudes were significantly larger in the case of an anticipated takeover as compared to a non-takeover situation ($p < 0.05$). In the analysis of ERSPs (Fig 2B), significant differences in alpha band over motor cortical areas prior to the experience of a robot takeover situation from human versus robot continuation were found ($p < 0.05$). In addition, in contrasting an anticipated human takeover from robot versus robot continuation (Fig 2C) significant differences in alpha band over motor cortical areas starting around 700 ms and significant differences in theta band over prefrontal cortical areas around 500 ms were found ($p < 0.05$). Lastly, in contrasting the human takeover from robot versus robot continuation significant differences were found in beta band over the left temporal cortex at several time points, but mostly pronounced around 600 ms ($p < 0.05$). An additional comparative analysis was performed, to investigate whether the effects observed in contrasting an anticipated human takeover from robot versus robot continuation (Fig 2C) are generalizable or specific to the two collaboration scenarios. Fig 3 shows the comparison between the two collaboration scenarios with Fig 3A depicting the grand average responses during the *sC*, and Fig 3B depicting the grand average responses during the *iC*. Although, the ERP and topographic plots look qualitatively comparable between the two scenarios, significant differences between conditions were only found for the *iC*. The analysis of ERSPs revealed significant differences in alpha band over motor cortical areas and significant differences in theta band over prefrontal cortical areas in both scenarios. However, the significant differences in motor-related activity are stronger and start earlier in *sC* as compared to *iC*, whereas the pre-frontal theta activity is more pronounced in *iC* as compared to *sC*.

**2.6.2 Single-trial decoding of anticipated takeover situations.** Table 2 shows the average ERP decoder performance across all subjects for each class of the two binary classification problems in the context of the two collaboration scenarios. Results are around the sample-size adjusted chance-level (determined via the inverse binomial test, slightly varied for each subject because of varying numbers of samples, but was on average $p_{chance}$ = 54.3%.) in all cases, except for the discrimination of human takeover from the robot ($RH_{tr-1}$) versus robot continuation ($RR_{tr-1}$). Here, the classification performance reaches $TNR_{RR}$ = 57.9±4.3% and $TPR_{RH}$ = 56.1 ±3.2% and an overall accuracy of $ACC_{RRvsRH}$ = 57.1±3.7% average across subjects. Detailed per subject results of decoding performance are reported in S3 Appendix.

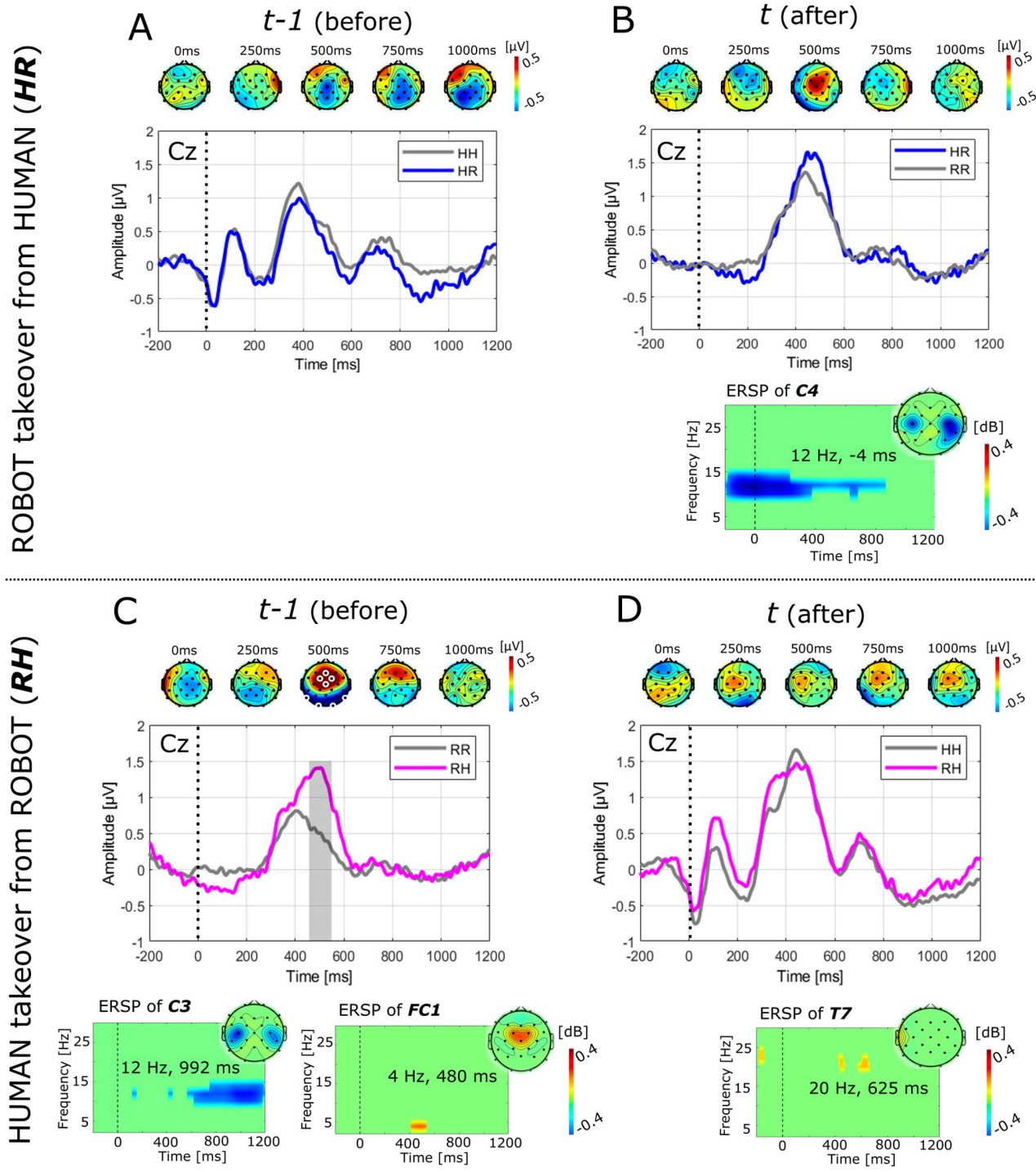

**Fig 2. ERP responses to human-robot takeover situations.** ERP time courses over channel Cz time-locked to the onset of robot movement before (*tr* − 1) and after (*tr*) an anticipated takeover situation average across all subject. Anticipated in this context refers to the user's mental preparation to the taking over action produced by the specific task allocation in the experimental protocol. All plots depict the grand average aggregating events of both collaboration scenarios (*sC*, *iC*). Plots in color (blue, magenta) depict ERP responses in takeover situations (*HR*, *RH*) and plots in grey depict ERP responses in non-takeover situations (*HH*, *RR*). Grey-shaded areas show significant time-points. The difference grand average is furthermore depicted as topographic plots at relevant time points above each plot, whereby white markers show significant electrodes in the respective time-point. The grand average difference ERSPs masked with the minimum (in blue) and maximum (in yellow-red) significant bins, are depicted below each plot (in Panel A, no ERSP results are reported because no significant bins were found). Panel A and B show the results of contrasting responses to a robot takeover from the human (*HR*) versus the human continuation (*HH*), before and after the anticipated takeover situation. Panel C and D show the results of

contrasting responses to a human takeover from the robot (*RH*) versus robot continuation (*RR*), before and after the alleged takeover situation. Subject specific ERP responses are detailed in S3 Appendix, Fig 1.

## 3 Simulation-based HRI feasibility study

### 3.1 Robot reinforcement learning model

A simulation-based study was conducted to explore the feasibility of a robot dynamically learning task-assignments from online decoded ERPs of the human interaction partner. The simulation assumes an interaction following the basic principles of the intermittent collaboration (*iC*) with two interaction partners, a human (*H*) and a robot (*R*), which perform a task together. This task consists of a set of subtasks *S*. From these $n$ subtasks *H* chooses $m$ subtasks which are unknown to *R*. *R*'s assigned subtasks are naturally the remaining subtasks which the human has not chosen. *R*'s goal is to learn which subtasks *H* has chosen and which subtasks *R* should take responsibility for. The robot chooses either to execute the subtask (robot action: $a_R$) or not (human action: $a_H$). This simulates the situation of human preference-based choice of subtasks or the situation of the human being physically or mentally unable to perform some of the subtasks, therefore choosing the remaining subtasks. The joint task between human and robot is modeled as a finite sequence of events *E* with each step of the sequence $e_k$ with $k \in \{1, \ldots, o\}$ requiring the execution of one of the subtasks $e_k \in S$. The simulated robot keeps a

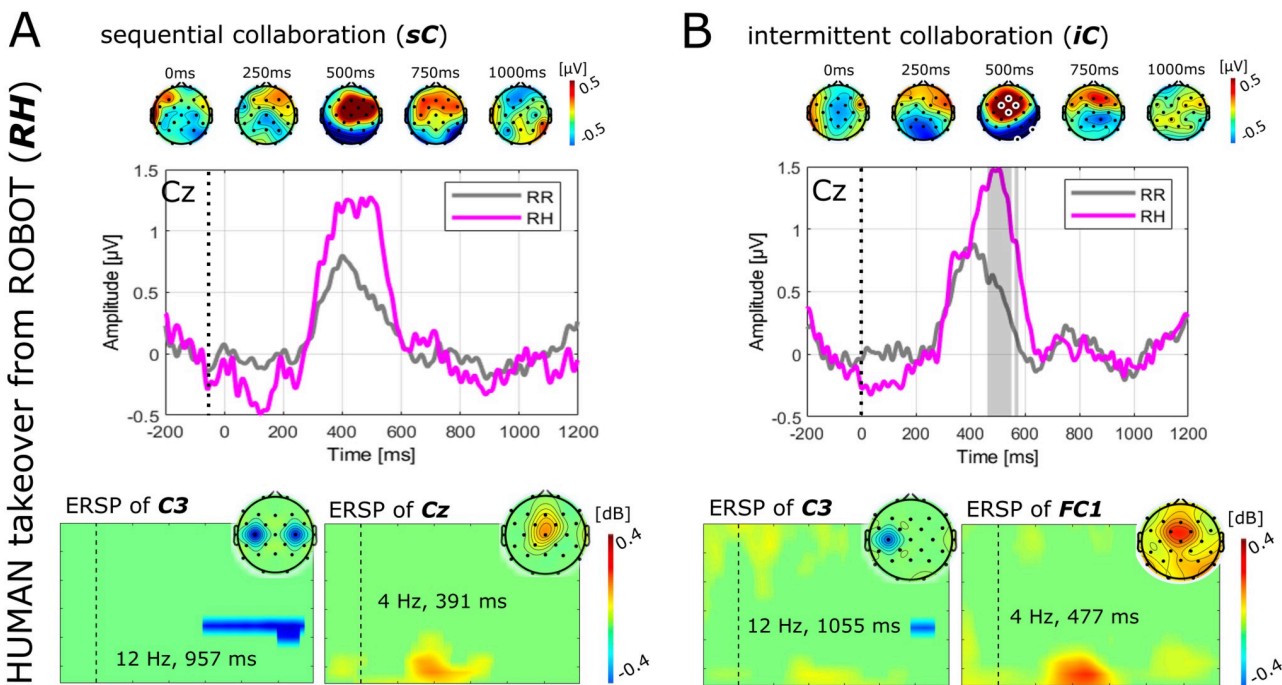

**Fig 3. ERP responses before an anticipated human-to-robot takeover comparing sequential and intermittent collaboration.** ERP time courses over channel Cz time-locked to the onset of robot movement comparing the anticipated human takeover from robot versus robot continuation across the two collaboration scenarios average across all subject. Panel A depicts the grand average responses during the sequential collaboration (*sC*), whereas panel B depicts the grand average responses during the intermittent collaboration (*iC*). The difference grand average is furthermore depicted as topographic plots at relevant time points above each plot. White-markings show significant electrodes in the respective time-point. In addition, the grand average difference ERSPs masked with the minimum (in blue) and maximum (in yellow-red) significant bins, are depicted below each plot. Subject specific ERP responses are detailed in S3 Appendix, Fig 1.

**Table 2. Results of single-trial decoding performance.** Average across subjects (n = 11) single-trial classification performance (M±SD [%]) for decoding anticipation of variants of takeover (*HR*, *RH*) against non-takeover situations (*HH*, *RR*) for sequential collaboration (*sC*) and intermittent collaboration (*iC*). Bolded numbers show results that are significantly above the sample-size adjusted chance-level $p_{chance} = 54.3\%$. Detailed per subject results of decoding performance are reported in S3 Appendix.

| | $TNR_{HH}$ | $TPR_{HR}$ | $ACC_{HHvsHR}$ | $TNR_{RR}$ | $TPR_{RH}$ | $ACC_{RRvsRH}$ |
|---|---|---|---|---|---|---|
| *sC* | 52.0±3.7 | 51.4±5.1 | 51.9±3.7 | 52.9±2.6 | 53.3±5.3 | 53.0±3.0 |
| *iC* | 51.5±3.4 | 51.4±3.5 | 51.4±3.3 | **57.9±4.3*** | **56.1±3.2*** | **57.1±3.7*** |

memory $q(s_i, a_j)$ with $s_i \in S$ and $a_j \in A = \{a_H, a_R\}$. This memory is initialized uniformly among all possible subtasks and actions with $q^{tr=0}(s_i, a_j) = 0$ for all $i, j$ and updated after each trial *tr* using a learning function Eq (3) inspired by reinforcement learning [80],

$$q^{tr}(s_i, \hat{a}) \leftarrow q^{tr}(s_i, \hat{a}) + \alpha R \tag{3}$$

with $\alpha$ being the learning rate, $R$ being a constant reward of 1 and $\hat{a}$ being the predicted next action (see Eq (4)). The anticipated responsibility for the next action $\hat{a}$ is derived from a simulated ERP decoder. This simulated ERP decoder was realized as a stochastic sampling from a uniform distribution $\mathcal{U}(0, 100)$ and comparison with the simulated ERP decoder rate. A random sample below the ERP decoder rate results in a correctly classified response, whereas a random sample above the ERP decoder rate results in a misclassified response. In accordance to the four cases defined in Table 1, four distinct ERP decoder rates ($TNR_{HH}$, $TPR_{HR}$, $TNR_{RR}$, $TPR_{RH}$) are used in the simulation. The decision for $\hat{a}$ is computed via Eq (4).

$$\epsilon \sim \mathcal{U}(0, 100)$$

$$\hat{a} = \begin{cases} a_H & \text{if } (a^{tr-1} = a_H \quad \wedge \quad a^{tr} = a_H) \quad \wedge \quad (\epsilon \leq TNR_{HH}), \quad \text{else} \quad a_R \\ a_R & \text{if } (a^{tr-1} = a_H \quad \wedge \quad a^{tr} = a_R) \quad \wedge \quad (\epsilon \leq TPR_{HR}), \quad \text{else} \quad a_H \\ a_R & \text{if } (a^{tr-1} = a_R \quad \wedge \quad a^{tr} = a_R) \quad \wedge \quad (\epsilon \leq TNR_{RR}), \quad \text{else} \quad a_H \\ a_H & \text{if } (a^{tr-1} = a_R \quad \wedge \quad a^{tr} = a_H) \quad \wedge \quad (\epsilon \leq TPR_{RH}), \quad \text{else} \quad a_R \end{cases} \tag{4}$$

Finally, the memory is turned into a probabilistic decision policy $\pi$ for more convenient reporting of the hypothetical decision accuracy using the softmax function Eq (5).

$$\pi^{tr} = p(a_j|s_i) = \frac{e^{q^{tr}(s_i, a_j)/\tau}}{\sum_{j=1}^{2} e^{q^{tr}(s_i, a_j)/\tau}} \tag{5}$$

The temperature parameter $\tau$ modulates how sensitive probabilities $p$ are to changes in values of $q$. For a low $\tau$ the probability of the highest value will be almost 1 and the others approximately zero. Whereas for large values of $\tau$ the probabilities are more smoothed out and more similar to each other, this way determining the speed and robustness of the learning [80]. Finally, after each trial update, the weights in the memory are standardized according to Eq (6) to avoid their values to increase towards very large numbers which would cause the divergence between choice options to gradually deflate.

$$q = \frac{q - \bar{q}}{SD(q)} \tag{6}$$

where $SD(q)$ represents the single standard deviation of $q$.

## 3.2 Simulations

Given the final model, the following four simulations (SIM1–4) were performed. Simulations SIM1–3 assume $n = 4$ subtasks out of which $m = 2$ are selected by $H$ unbeknownst to $R$. In SIM4, the number of subtasks and balance of subtask assignment is varied. For all simulations, the model parameters were fixed and manually set to $\alpha = 0.05$ and $\tau = 0.1$, as these values led to a good trade-off between learning speed and robustness, e.g. steep and smooth learning curves. All simulations assume an episode of length $o = 10$ as we believe this to be a realistic number for a real world product assembly task. Episodes were derived by sampling $o$ random choices from the available subtasks. The human choice of subtasks is derived by sampling $m$ random choices from the available subtasks.

**SIM1. Feasibility with realistic ERP decoder performance** Upon start of the simulation, the memory is initialized uniformly with $q^{tr=0}(s_i, a_j) = 0$ for all $s_i, a_j$, a subtask-sequence is derived, and the human choice of subtasks is derived. The memory is then updated in each new trial via Eq (3) over the course of 200 episodes. The decoding performances $TNR_{HH}$, $TPR_{HR}$, $TNR_{RR}$, $TPR_{RH}$ utilized in Eq (4) are set to the empirically determined average numbers derived from the intermittent collaboration ($iC$) of the experimental study (see results in Section 2.6.2). To obtain enough simulated data for allowing statistical analyses, above described procedure is repeated for 100 times with re-initialization of $q^{tr=0}(s_i, a_j) = 0$, re-initialization of the sequence, and re-initialization of the human choice of subtasks, in each repetition. This simulates a complete re-start of learning in each repetition. Finally, the average learning curve across repetitions is computed and reported as result.

**SIM2. Hypothetical variations of ERP decoder performance** The SIM2 procedure is similar to SIM1's, however, the ERP decoder rates $TNR_{HH}$, $TPR_{HR}$, $TNR_{RR}$, $TPR_{RH}$ in Eq (4) are compared between the empirically determined results from the experimental study and a set of hypothetical values consisting of four levels, from chance-level to a realistic upper bound: 50%, 60%, 70%, 80%. For simplicity, all four individual ERP decoder rates are set to the same value in the simulation of the above specified hypothetical levels. As in SIM1, the procedure is repeated for 100 times for each level and finally, the average learning curves across repetitions are computed per level and reported as results.

**SIM3. Dynamic re-learning of subtask-assignment** The SIM3 procedure is similar to the procedure in SIM2. After 40 episodes, $H$ would choose a new set of $m = 2$ subtasks and $R$'s task is to re-learn the subtask assignment by adapting its memory $q$. Critically, the moment of re-assignment of subtasks through $H$ is unknown to $R$, such that no re-initialization of $q$ is possible, but rather $q$ needs to be overwritten. This simulates the situation of the human partner dynamically changing preferences for subtasks during the collaboration, without $R$ knowing about when this re-selection of subtasks happened. As in SIM2, the procedure is repeated for 100 times for each level and finally, the average learning curves across repetitions are computed and reported as results.

**SIM4. Scalability to more subtasks** The SIM4 procedure is also similar to the procedure in SIM2, however, it includes variations of the number of subtasks $n$ and subtask-assignment $m$. The following test cases were chosen for the simulation: $n = 6$ and $m = 3$; $n = 10$ and $m = 5$; $n = 10$ and $m = 2$. The first two test cases simulate a gradual increase of the number of subtasks with keeping the balance between $H$ and $R$ subtask- assignment. The last case simulates an imbalanced assignment of subtasks where $H$ selected two out of ten subtasks. These simulations explore both the scalability of the approach, as well as its robustness towards imbalanced subtask-assignment. As in SIM2, the procedure is repeated for 100 times for each level and finally, the average learning curves across repetitions are computed and reported as results.

## 3.3 Results of simulation-based HRI feasibility study

The simulation results are depicted in Fig 4. Fig 4A shows the results of SIM1 (see Section SIM1). Despite the low ERP decoding performance of anticipated takeover, the simulated robot subtask choice accuracy reaches on average 70% around 35 episodes and 80% around 100 episodes. By assuming a single trial to have a duration of 2 seconds, this corresponds to around 6 min or 17 min respectively (note that the duration of one trial was approximately 2 seconds in the experimental HRI study). Note that SIM1 is based on decoding parameters

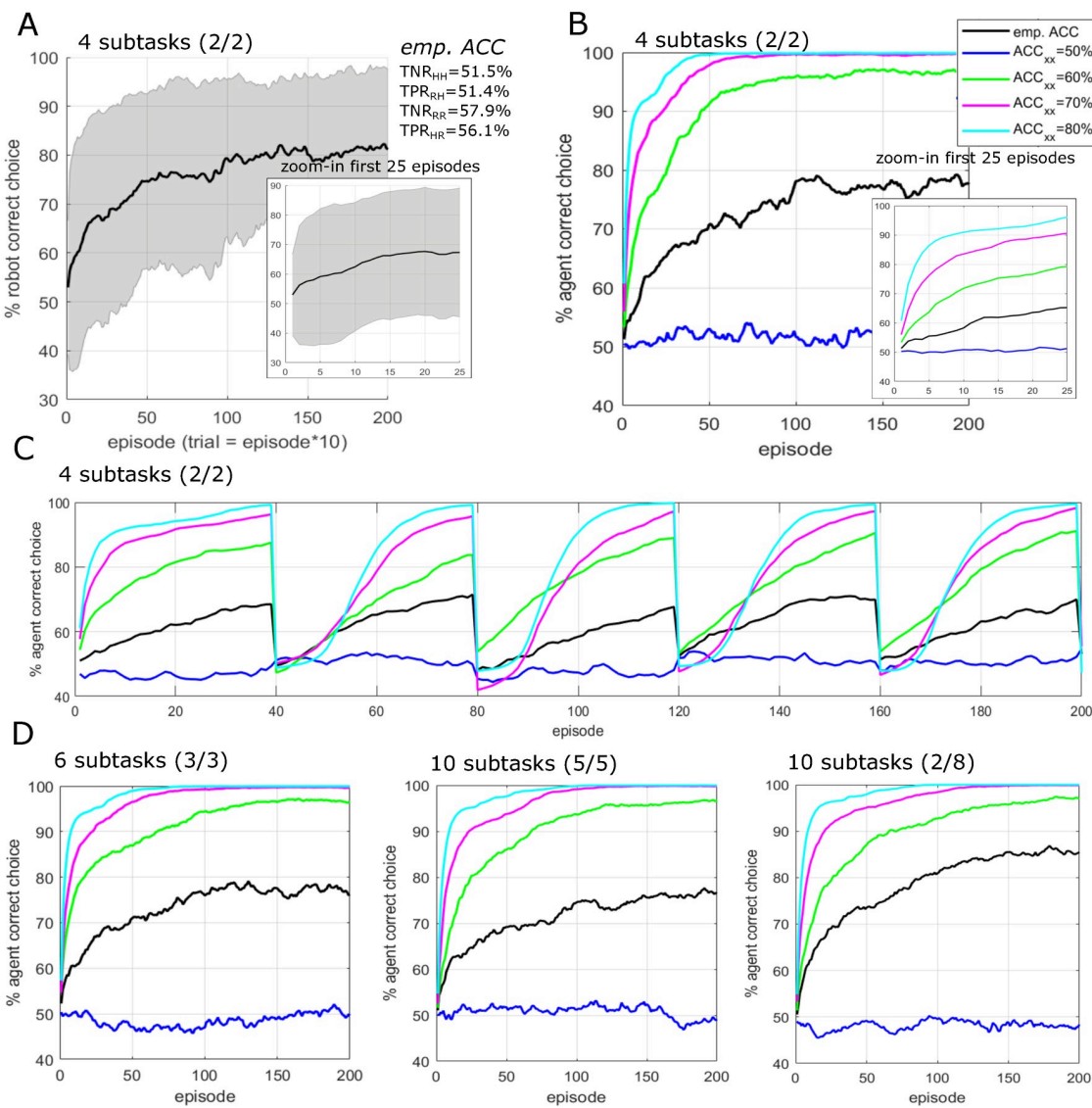

**Fig 4. Results of simulation-based HRI feasibility study.** Panel A: robot learning curve for decoding performance in line with the empirically determined (*emp. ACC*) results of the experimental study. Panel B: various robot learning curves for the empirical decoding performance (*emp. ACC*) and hypothetical variations of ERP decoding performance of anticipated takeover (whereby $ACC_{xx}=TNR_{HH}=TPR_{RH}=TNR_{RR}=TPR_{RH}$). Panel C: learning curves in case of a dynamic task re-assignment after every 400 trials / 40 episodes. Panel D: variations of the original experiment with increased numbers of subtasks illustrating the scalability of the approach towards more complex tasks as well as robustness towards imbalanced subtask assignment (6 subtasks with balanced assignement, 10 subtasks with balanced assignment, and 10 subtasks with 2 human and 8 robot assigned subtasks). In the example chosen for the simulation, a single episode corresponds to 10 trials.

(further denoted as *empirical decoding performance*) matching those empirically determined during our experimental study. Therefore, the results of SIM1 provide the closest estimate of how the robot reinforcement learning approach would perform in a real HRI experiment. SIM2 cross-compares the the empirical decoding performance with hypothetical decoding performances that are lower and higher than the empirical decoding performance. Fig 4B shows the results of SIM2 (see Section SIM2). The results indicate that higher ERP decoding performance results in faster learning, but also that successful learning is feasible as long as decoding performance is above chance-level (compare 50% with the empirical decoding performance). Fig 4C shows the results of SIM3 (see Section SIM3) in which the case of a hypothetical task re-assignment after every 40 episodes was simulated, e.g. the human partner choosing a new set of tasks every 6–7 min. Results indicate successful re-learning of task assignment without notable loss of choice accuracy in the long run. Fig 4D shows the results of SIM4 (see Section SIM4) in which hypothetical variations of the original experiment with increased numbers of subtasks, e.g. 6 or 10 subtasks, was simulated. The results show scalability of the approach towards more complex tasks, as well as robustness toward imbalanced subtask assignment. Specifically, the results show that more subtasks generally result in decreased learning speed. Imbalanced subtask assignment does not compromise the robot learning, e.g. in the case when the human chooses less or more than half of the available subtasks, the subtask-assignement is learned by the robot with comparable performance. The current implementation of the robot learning algorithm results in steeper robot learning curves when more subtasks are assigned to the robot compared to the human and vice-versa more shallow learning curves when less subtasks are assigned to the robot (see example in the rightmost plot in Panel (d): human choosing 2 subtasks and leaving 8 subtasks to the robot).

## 4 Discussion

### 4.1 A neural correlate of anticipation of human-robot takeover?

The results of our study revealed several spatio-temporal and temporo-spectral components in the contrasts of the various studied human-robot takeover situations. Given the latencies observed for significant differences in the ERP and the spectral and spatial distribution of the differences observed in the ERSP these components can be associated to both attentive/cognitive and motor-related brain processes. For instance, the event-related desynchronization of alpha band power in Fig 2B and 2C is likely related to differential levels of preceding or subsequent motor preparation and planning. Specifically, Fig 2B shows the contrast of a preceding motor action versus idling (e.g. no action from the human required) and Fig 2C a subsequent motor action versus idling. The beta activity observed over left temporal sites (see Fig 2D) could be related to a stronger beta rebound in the case of a preceding takeover situation as the takeover situation likely involved increased engagement of motor-related activity (where the next action was difficult for the subject to anticipate) as compared to a non-takeover situation (where the next action could be better anticipated). These observations are likely related to anticipatory movement-related brain processes involved in motor planning and preparation (see for instance [81–84]). Besides movement-related anticipatory activities, a significant component in the ERP over fronto-central sites at latencies around 500 ms post onset of robot movement was observed when contrasting a human takeover from the robot versus the robot continuation. The effect was found to be significant both in the spatio-temporal and the time-frequency analysis, which indicates that the observed effect was both time-and phase-locked to the onset of the robot motion. The observed latency and spatial location suggests the effect to be a P300 component modulation related to an attention re-orientation due to a "match/mismatch with a consciously-maintained working memory trace" [85], e.g. a "surprise" signal

indicating the subject to re-focus on the task after realizing that it is now his/her turn. This interpretation is further supported by the observed differences between the two collaboration scenarios (Fig 3) since subjects likely anticipated takeover situations in different ways for each of the two scenarios: in the sequential collaboration, subjects could possibly anticipate the take-over situation already several steps/actions beforehand, since the moment of reaching the boundary was quite obvious. Whereas in the intermittent collaboration, takeover situations were less obvious, therefore involving stronger activities related to attention re-orientation. In summary, our findings indeed represent anticipatory brain activity that putatively originate from both attention- and motor-related brain processes. Whether both effects are functionally independent from each other, e.g whether the ERP can be observed in the absence of a subsequent motor action requires further investigation in follow-up studies.

## 4.2 Decoding anticipated takeover situations from single-trial EEG

The results from the decoding performance evaluation provide evidence that the discrimination between an anticipated human takeover from a robot ($RH_{tr-1}$) and robot continuation ($RR_{tr-1}$), can be correctly classified with accuracies above the sample-size adjusted chance-level for the intermittent collaboration. This discriminator represents two situations: 1) when humans anticipate a takeover action from human to robot; and 2) when humans expect the robot to continue with the task. These situations indicate that it is possible to identify when the user is willing to accept the robot to continue performing the task, or when the user prefers to execute the next action. These important events are two essential elements for the correct task planning in HRI scenarios. In particular, they can be used to adapt the task plan to the user's preferences, identifying tasks assigned to the robot that the user expects to execute. However, while the anticipation of the human taking over from the robot ($RH$) could be classified above chance, the anticipation of a robot taking over from the human ($HR$) could not be properly classified. While the current results demonstrate that even low performant decodeability of HRI takeover situations can lead to effective robot learning of subtask assignment, further research is required to be able to additionally decode the anticipation of a robot taking over from the human ($HR$), which would be highly beneficial, since the human anticipation of a robot taking over implicitly indicates that the human expects robot assistance. Such situations are important to identify in order to enable correct task allocations. In particular, it can be used to directly trigger the robot assistance in the task planner.

## 4.3 Robot learning from feedback based on anticipatory measures

Utilizing feedback based on anticipatory measures for robot learning bears an intriguing advantage over using reactive/evaluative measures, such as the ErrP. Using information about anticipated events allows for the possibility that actual events (those that are anticipated) will not necessarily need to take place. The robot can learn from the feedback derived from EEG responsed due to anticipations without the need to perform actions that lead to decodeable responses such as in the case of the ErrP which is exclusively elicited when the human is observing and mentally evaluating a robot's action. In a future HRI scenario, this would enable the robot, for instance, to collaborate with the human in a "semi-closed-loop" fashion: For instance, the robot could initially collaborate with the user using a default policy and learn silently in the background from the user's responses to anticipated event. Subsequently, when enough information in the form of classifier decisions have been collected, the robot could swap the default policy for the internally learned/adapted policy, which includes the user's preferences. The learning algorithm proposed in the simulation study investigated exactly this adaptation scheme, e.g. unidirectional robot learning of the user preferred task-assignment "in

the background". The results of the simulation study showed that despite the relative low decoding performance, successful learning and re-learning of task-assignments between human and robot is feasible. This is because of the exploitation of the long-term effects of reinforcement learning. The learning function accumulates the information provided by individual classifier decisions into a converging policy, thereby accommodating, e.g. averaging across the noisy/imprecise feedback provided by single decoded ERPs. One important aspect in the learning algorithm is the per-trial-standardization of the weights (see Eq 6), which enables the dynamic re-learning of subtask-assignment in situations when the user is changing subtask preference during collaboration. A scalability analysis up to 10 subtasks (in comparison to 4 subtasks) revealed that the approach is well scalable to more complex HRI with the only consequence of a flattened learning curve, e.g. the learning of subtask-assignment taking longer than with less number of subtasks. Furthermore, imbalanced assignment of subtasks does not affect the learning, e.g. it does not matter if the human selected more or less subtasks than the robot; the robot will learn to choose the remaining tasks, no matter how many. These results indicate that using the information provided by the decoding ERPs alone, a robot can flexibly identify (learn) the tasks a user prefers to execute, most importantly without including any other information overtly accessible to the robot sensors. A rather logical outcome of the simulation was the intrinsic correlation of the ERP decoding performance with the robot's learning accuracy and speed—the less noisy the human feedback, the more precise and faster the robot learning. This renders the need for better decoding performance, an important aspect to explore in future work. In this regard, it is important to emphasize that in the present work only ERP time-domain features were used for the classification. These features mostly reflect low frequency components in the EEG signal. This was intended, since the differential motor activity was not in the center of our attention. Depending on the scope of future research, an extension of the decoding approach may include (time-)frequency domain components as complementary features for improving decoding performance and thus speeding up robot learning.

### 4.4 Outlook on combining anticipatory and evaluative measures

An intriguing possibility to further enhance the information provided to the robot is the combination with other neuro-cognitive measures previously reported and investigated in the context of augmenting HRI. For instance, it is conceivable to combine anticipatory measures, such as the ERP observed in our study and the movement-related measures with reactive/evaluative measures, such as the well-studied ErrP [57, 60–62, 64, 75, 86, 87] and related measures of error- and performance monitoring [10, 88, 89]. Such combination might be useful either to re-evaluate or to augment the previously harvested information decoded from anticipatory activity. Furthermore, it could be useful to combine these single-trial measures with long-term measures of human trust towards a machine/robot during interaction which have been shown to be decodeable from fMRI activity [30, 90] and more recently also from EEG signals [10, 31, 32, 91].

### 5 Conclusions

The present work explored the feasibility of using EEG-based neuro-cognitive measures for online robot learning of dynamic subtask assignment. To this end, we demonstrated in an experimental human subject study the presence of EEG measures indicative of the human partner anticipating a takeover situation from the human to the robot or vice-versa. This response could be classified in single-trial with an average accuracy of 57.1±3.7% across subjects. This work further proposes a reinforcement learning based algorithm employing these measures as a neuronal feedback signal from the human to the robot for dynamic learning of

subtask-assignment. Simulation-based studies demonstrated successful and stable learning of robot subtask-assignment despite low decoding performances. The simulations further showed that dynamic re-learning of subtask-assignment is feasible and misbalanced assignment of subtasks does not affect the learning success. These findings demonstrate the usability of EEG-based neuro-cognitive measures to mediate the complex and largely unsolved problem of human-robot collaborative task planning.

## Supporting information

**S1 Appendix. Validation of robot motion timing.**
(PDF)

**S2 Appendix. Statistical testing.**
(PDF)

**S3 Appendix. Per subject results of ERP responses and single-trial classification performance.**
(PDF)

## Author Contributions

**Conceptualization:** Stefan K. Ehrlich, Emmanuel Dean-Leon, Viktorija Dimova-Edeleva, Gordon Cheng.

**Data curation:** Stefan K. Ehrlich.

**Formal analysis:** Stefan K. Ehrlich.

**Investigation:** Stefan K. Ehrlich, Emmanuel Dean-Leon, Nicholas Tacca, Simon Armleder, Viktorija Dimova-Edeleva.

**Methodology:** Stefan K. Ehrlich, Emmanuel Dean-Leon, Nicholas Tacca, Simon Armleder.

**Project administration:** Gordon Cheng.

**Software:** Stefan K. Ehrlich, Emmanuel Dean-Leon, Nicholas Tacca, Simon Armleder.

**Supervision:** Gordon Cheng.

**Validation:** Stefan K. Ehrlich, Emmanuel Dean-Leon, Gordon Cheng.

**Visualization:** Stefan K. Ehrlich.

**Writing – original draft:** Stefan K. Ehrlich, Emmanuel Dean-Leon, Nicholas Tacca, Simon Armleder, Viktorija Dimova-Edeleva.

**Writing – review & editing:** Stefan K. Ehrlich, Emmanuel Dean-Leon, Nicholas Tacca, Simon Armleder, Viktorija Dimova-Edeleva, Gordon Cheng.

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
