## [Decision Letter · Decision Letter 0]

17 Jun 2022

PONE-D-21-32176Neuro-cognitive measures of anticipated takeover situations enable human-robot collaborative task planningPLOS ONE

Dear Dr. Ehrlich,

Thank you for submitting your manuscript to PLOS ONE. After careful consideration, we feel that it has merit, however, requires revisions to add clarifications and additional information as indicated by the reviews below. Therefore, we invite you to submit a revised version of the manuscript that addresses the points raised during the review process. Please submit your revised manuscript by Aug 01 2022 11:59PM. If you will need more time than this to complete your revisions, please reply to this message or contact the journal office at plosone@plos.org. Please include the following items when submitting your revised manuscript:A rebuttal letter that responds to each point raised by the academic editor and reviewer(s). You should upload this letter as a separate file labeled 'Response to Reviewers'.A marked-up copy of your manuscript that highlights changes made to the original version. You should upload this as a separate file labeled 'Revised Manuscript with Track Changes'.An unmarked version of your revised paper without tracked changes. You should upload this as a separate file labeled 'Manuscript'.If applicable, we recommend that you deposit your laboratory protocols in protocols.io to enhance the reproducibility of your results. Protocols.io assigns your protocol its own identifier (DOI) so that it can be cited independently in the future. For instructions see: https://journals.plos.org/plosone/s/submission-guidelines#loc-laboratory-protocols. Additionally, PLOS ONE offers an option for publishing peer-reviewed Lab Protocol articles, which describe protocols hosted on protocols.io. Read more information on sharing protocols at https://plos.org/protocols?utm_medium=editorial-email&utm_source=authorletters&utm_campaign=protocols.

We look forward to receiving your revised manuscript.

Kind regards,

Hasan Ayaz, PhD

Academic Editor

PLOS ONE

**Journal requirements:**

2. Please provide additional details regarding participant consent. In the Methods section,

please ensure that you have specified (1) whether consent was informed and (2) what type you obtained

(for instance, written or verbal). If your study included minors, state whether you obtained consent from parents

 or guardians. If the need for consent was waived by the ethics committee, please include this information.

“This work was partially supported by the Elite Network Bavaria (ENB) through the 793

master program in neuroengineering (MSNE)”

Reviewers' comments:

Reviewer's Responses to Questions

**Comments to the Author**

1. Is the manuscript technically sound, and do the data support the conclusions?

Reviewer #1: Partly

Reviewer #2: Yes

2. Has the statistical analysis been performed appropriately and rigorously? 

Reviewer #1: Yes

Reviewer #2: Yes

3. Have the authors made all data underlying the findings in their manuscript fully available?

Reviewer #1: Yes

Reviewer #2: Yes

4. Is the manuscript presented in an intelligible fashion and written in standard English?

Reviewer #1: Yes

Reviewer #2: Yes

5. Review Comments to the Author

Reviewer #1: The authors reflect knowledge in the technical aspects of robot control, data analysis and manuscript writing but seem to miss a clear aspect of the signals studied in the context of motor control: movement preparation or anticipatory activity. For this reason, the data segmentation and its posterior analysis becomes less solid.

Overall, the authors need to reconsider the data segmentation and rerun their analysis to truly describe the activity prior to the takeover. There is a body of research in the field of movement preparation, anticipatory tasks, and human-robot collaboration that the authors can use to better inform their future data segmentation and analysis, which is necessary for the manuscript to be publishable.

See the attached file for more details.

Reviewer #2: I enjoyed this paper and I think it will be an important contribution. I would recommend to accept with minor revisions.

I have three comments / suggestions:

* I read the task description and reviewed the figures on the task. It’s still unclear to me exactly what the task is and how collaboration and hand-offs happened between the human and the robot. I would suggest to add a lot more detail about this so future researchers can accurately reproduce this work or approximate tasks like it.

* The paper emphasizes it is unique because it uses anticipatory human EEG signals to guide human-robot collaboration. Previous research (see below) has focussed on using the occurrence of errors to guide behavior and how this impacts construct like human-robot and human-automation trust. In that research, the error signals associated with the discrepancy between expected and actual behavior is what is most predictive and could be used to guide machine behavior in a neuroadaptive system. However, given the results presented in this paper, this raises the intriguing possibility that several signals (anticipatory and reactive) could be combined to improve HRI. For example, earlier fMRI work suggested that an “intention to trust” signal moved from a reactionary signal to an anticipatory one (King-Casas et al., 2005). Using the literature below, I would suggest to add a point (the relationship between anticipatory and reactive signals) like this in the discussion.

* [neuroadaptive systems] Zander, T. O., Krol, L. R., Birbaumer, N. P., & Gramann, K. (2016). Neuroadaptive technology enables implicit cursor control based on medial prefrontal cortex activity. Proceedings of the National Academy of Sciences, 113(52), 14898-14903

* [trust] King-Casas, B., Tomlin, D., Anen, C., Camerer, C. F., Quartz, S. R., & Montague, P. R. (2005). Getting to know you: reputation and trust in a two-person economic exchange. Science, 308(5718), 78-83

* [monitoring automation] - Berberian, B., Somon, B., Sahaï, A., & Gouraud, J. (2017). The out-of-the-loop Brain: a neuroergonomic approach of the human automation interaction. Annual Reviews in Control, 44, 303-315

* [human vs automation] - Somon, B., Campagne, A., Delorme, A., & Berberian, B. (2019). Human or not human? Performance monitoring ERPs during human agent and machine supervision. Neuroimage, 186, 266-277

* [detecting error] - Fedota, J. R., & Parasuraman, R. (2010). Neuroergonomics and human error. Theoretical Issues in Ergonomics Science, 11(5), 402-421

* [monitoring automation / trust] - De Visser, E. J., Beatty, P. J., Estepp, J. R., Kohn, S., Abubshait, A., Fedota, J. R., & McDonald, C. G. (2018). Learning from the slips of others: Neural correlates of trust in automated agents. Frontiers in human neuroscience, 12, 309

* [detecting error] Weller, L., Schwarz, K. A., Kunde, W., & Pfister, R. (2018). My mistake? Enhanced error processing for commanded compared to passively observed actions. Psychophysiology, 55(6), e13057

* [trust] Akash, K., Hu, W. L., Jain, N., & Reid, T. (2018). A classification model for sensing human trust in machines using EEG and GSR. ACM Transactions on Interactive Intelligent Systems (TiiS), 8(4), 1-20

* [trust] Wang, M., Hussein, A., Rojas, R. F., Shafi, K., & Abbass, H. A. (2018, November). EEG-based neural correlates of trust in human-autonomy interaction. In 2018 IEEE Symposium Series on Computational Intelligence (SSCI) (pp. 350-357). IEEE

* [trust] Choo, S., & Nam, C. S. (2022). Detecting Human Trust Calibration in Automation: A Convolutional Neural Network Approach. IEEE Transactions on Human-Machine Systems

* [trust] Goodyear, K., Parasuraman, R., Chernyak, S., de Visser, E., Madhavan, P., Deshpande, G., & Krueger, F. (2017). An fMRI and effective connectivity study investigating miss errors during advice utilization from human and machine agents. Social neuroscience, 12(5), 570-581

* [trust] Goodyear, K., Parasuraman, R., Chernyak, S., Madhavan, P., Deshpande, G., & Krueger, F. (2016). Advice taking from humans and machines: An fMRI and effective connectivity study. Frontiers in Human Neuroscience, 10, 542

* I would suggest to add the term “Human-Robot Interaction” to your work in the abstract or elsewhere in addition to the term HRC. The way you describe HRC is pretty much the description of the HRI field. HRC is more of a subcategory that describes how exactly human/robots/machines/automation should work together and coordinate, a subject of inquiry in both the HRI and human factors fields. This HRI term refers to the entire field / discipline (see HRI conference for example) and will make it so your article has broader appeal to the HRI community as well as the neuroscience communities and the neuroergonomic community.

* In addition some relevant literature from human factors / HRI:

* [HRI and neuroscience] Henschel, A., Hortensius, R., & Cross, E. S. (2020). Social cognition in the age of human–robot interaction. Trends in Neurosciences, 43(6), 373-384

* [transparency] Chen, J. Y., Lakhmani, S. G., Stowers, K., Selkowitz, A. R., Wright, J. L., & Barnes, M. (2018). Situation awareness-based agent transparency and human-autonomy teaming effectiveness. Theoretical issues in ergonomics science, 19(3), 259-282

* [transparency] Mercado, J. E., Rupp, M. A., Chen, J. Y., Barnes, M. J., Barber, D., & Procci, K. (2016). Intelligent agent transparency in human–agent teaming for Multi-UxV management. Human factors, 58(3), 401-415

* [Function allocation] De Winter, J. C., & Dodou, D. (2014). Why the Fitts list has persisted throughout the history of function allocation. Cognition, Technology & Work, 16(1), 1-11

* [Function allocation] Kaber, D. B. (2018). Issues in human–automation interaction modeling: Presumptive aspects of frameworks of types and levels of automation. Journal of Cognitive Engineering and Decision Making, 12(1), 7-24

* [adaptive automation] Parasuraman, R., Bahri, T., Deaton, J. E., Morrison, J. G., & Barnes, M. (1992). Theory and design of adaptive automation in aviation systems. Catholic Univ of America Washington DC cognitive science lab

* [adaptive automation] Scerbo, M. (2007). Adaptive automation. Neuroergonomics: The brain at work, 239252

* [adaptive automation] Parasuraman, R., Mouloua, M., Molloy, R., & Hilburn, B. (1993, May). Adaptive function allocation reduces performance cost of static automation. In 7th international symposium on aviation psychology (pp. 37-42)

6. PLOS authors have the option to publish the peer review history of their article (what does this mean?). If published, this will include your full peer review and any attached files.

Reviewer #1: **Yes: **André F. Salazar-Gómez

Reviewer #2: No

---

## [Author Response · Author response to Decision Letter 0]

10 Nov 2022

Dear Editor and reviewers,

We thank the editor and reviewers for their constructive and insightful comments and recommendations which have helped to significantly improve the manuscript. We have revised the manuscript according to all recommendations and carried out the corrections as suggested. Please find individual replies to each of the editor’s and reviewer’s comments including references to the respective sections in our revised manuscript. All changes to the manuscript are indicated in the responses and marked in blue in the ‘Revised Manuscript with Track Changes’. Please note that line number references are valid only for the ‘Revised Manuscript with Track Changes’. A separate clean version without tracked changes, named ‘Manuscript’, also accompanies this revision. 

Reviewer 1:

The authors reflect knowledge in the technical aspects of robot control, data analysis and manuscript writing but seem to miss a clear aspect of the signals studied in the context of motor control: movement preparation or anticipatory activity. For this reason, the data segmentation and its posterior analysis becomes less solid. Overall, the authors need to reconsider the data segmentation and rerun their analysis to truly describe the activity prior to the takeover. There is a body of research in the field of movement preparation, anticipatory tasks, and human-robot collaboration that the authors can use to better inform their future data segmentation and analysis, which is necessary for the manuscript to be publishable. See the attached file for more details.

Thank you very much for your submissions and for a clearly written paper. Overall, the paradigm proposed, and the technical approach has very positive approach. Nevertheless, there are three key aspects that need attention: the 1) neural signal explored, which involves a redefinition of the data segmented, 2) details about the methods and the protocol employed, and 3) the interpretation of the signal studied and the results. I do have a list of comments, I am sure most of them are easily solved by adding clarifications. Finally, the authors present an analysis of the signals in the context of motor control at the end of the document that is on spot. This should further support the need for the authors to explore motor preparatory activity as a driver for their study. 

General comment 1: Regarding the neural signal explored, it is not clear to me what the authors seek by analyzing ErrP activity in the context of anticipatory takeover. This is in part due to the absence of some key information regarding the experimental protocol (please see note about not finding any information regarding reference 60) but also due to how data is segmented prior to being analyzed (as defined in table 1). The use of ErrPs is intended for goal-oriented tasks, where errors are to be detected. Nevertheless, in the current paradigm, the authors do not define clearly errors guiding the subjects’ behavior. Moreover, most of the analysis in the paper (including the title) seem to focus on anticipatory behavior (as stated in the abstract: “prediction of the human anticipation”), but no anticipatory EEG activity (EEG measures of anticipatory behavior, mu rhythm, motor preparation, or anticipatory slow cortical potentials) is covered in the introduction, nor used to set the paradigm, extract data, analyze the results, discuss then and get to conclusions. This is a key element that need attention. 

The authors are encouraged to explore the concept of movement preparation and EEG measures of anticipatory behavior, which seem to connect seamlessly with anticipatory takeover and the paradigm presented in this manuscript. Consider this: To truly analyze an anticipated takeover situation, it could be useful to focus not on a comparison between trial t and trial t+1 but rather on the activity prior to the takeover event (prior to the robot onset due to a human button press). Motor anticipatory activity is expected to be present when a human will perform an action, and hence, this will be observed in the EEG activity some hundreds of milliseconds prior to the button press/robot movement onset.

There is a body of research in the field of EEG measures of anticipatory behavior and movement preparation (including the mu rhythm). If overall, the authors wish to focus on ErrPs, it is important to make clear for the reader, in the title, abstract and main body of the manuscript, why these are the signals explored and not preparatory/anticipatory rhythms. Please see the following citations for more info about movement preparation and anticipation: 

• M. Rodrigo, L. Montesano and J. Minguez, "Classification of resting, anticipation and movement states in self-initiated arm movements for EEG brain computer interfaces," 2011 Annual International Conference of the IEEE Engineering in Medicine and Biology Society, 2011, pp. 6285-6288, doi: 10.1109/IEMBS.2011.6091551.

Response: We thank the reviewer for this insightful comment and the recommendation to include literature covering anticipatory motor related EEG measures. We agree with the reviewer that previous works on anticipatory movement related EEG measures are an important part that we indeed missed to put in relation to our work which we have now thoroughly revised. In addition, we understand that the reviewer’s concerns were further driven by various missing details in the initial manuscript, specifically (i) the emphasis on ErrPs, both in the introduction, as well as the interpretation of the observed signal in the discussion, (ii) missing details about the experiment design and protocol, (iii) the use of the terms “trial t” and “trial t+1” might have been misleading, and (iv) the various locations in the initial manuscript where we stated our work to be the first of its kind. We have thoroughly revised the manuscript in light with this general and more specific comments below. We took the following measures to correct the manuscript:

- We added a paragraph on anticipatory movement related potentials and their use in BCI and likewise de-emphasized the details on ErrPs in the introduction, specifically in section “Neuroengineering approaches to HRI” (lines 129-150). Specifically, we replaced the previous paragraph on ErrPs with the following paragraph: “Other works have focused on anticipation- and error-related EEG signals. While the former is a motor preparatory signal prior to movement onset which is manifested as a slow negative wave over fronto-central areas in anticipation of an event, the latter is a phenomenon related to error- and performance monitoring (Ullsperger et al. 2014), observable upon human observation of an erroneous or unexpected event and manifested in a specific event-related potential (ERP) following the observed event (Falkenstein et al. 1991, Gehring et al. 1993, Miltner et al. 1997). Both phenomena have been studied in the context of BCI and HRI. Anticipatory movement related potentials have been shown to be decodeable in single trial in anticipation of upcoming events and demonstrated to be useful to improve BCIs or HRI (Gangadhar et al. 2008, Garipelli et al. 2011, Rodrigo et al. 2011, Chavarriaga et al. 2012, Garipelli et al. 2013). The error-related potential (ErrP) is evaluative in its nature, e.g. arises as a response to an observed action, independent of whether this action was self-inflicted or executed by an external instance, such as a robot. ErrPs have been proposed and successfully validated as a useful measure for the implicit assessment of erroneous or unexpected robot actions from the viewpoint of the human partner (Salazar-Gomez et al. 2017, Welke et al. 2017, Ehrlich & Cheng 2019) and demonstrated as an implicit human feedback signal for robot adaptation during HRI. ErrP-based robot adaptation was shown in several case studies, such as in the context of robot learning of a human desired end-effector trajectory by Iturrate and colleagues (Iturrate et al. 2015), a binary sorting task by Salazar-Gomez and colleagues (Salazar-Gomez et al. 2017), the recognition and replication of human gestures by Kim and colleagues (Kim et al. 2017), mediation of human-robot co-adaptation by Ehrlich & Cheng (Ehrlich & Cheng 2018), the inference of subject preferences on robot trajectories by Iwane and colleagues (Iwane et al. 2019), and very recently an example of robot learning control-policies for singularity avoidance by Aldini and colleagues (Aldini et al. 2021).”

- We removed the paragraph stating our study being the first of its kind to propose and investigate anticipatory EEG measures for HRI in section “Aim of this work” (lines 200-212).

- We thoroughly revised the descriptions of the experimental paradigm, task, and protocol (lines 235-279).

- We changed the terms “trial t” and “trial t+1” to “trial tr-1” and “trial tr” (to address specific comment 17, also) to reflect better in the terminology that the activity prior to the takeover situation was evaluated. This was changed throughout the manuscript, specifically in Table 1, Figure 2, and the various locations where it appeared in text.

- We thoroughly revised the discussion by restructuring the content, removing the discussion on signal clarity and experimental design, elaborated the interpretation of our observed effects by including anticipatory movement related EEG measures (lines 691-809), and added a discussion about combining anticipatory with reactive/evaluative measures to address reviewer 2, comment 2 (lines 798-809).

- Added relevant references suggested by the reviewer:

o Chavarriaga, R., Perrin, X., Siegwart, R., & Millán, J. D. R. (2012, January). Anticipation-and error-related EEG signals during realistic human-machine interaction: A study on visual and tactile feedback. In 2012 Annual International Conference of the IEEE Engineering in Medicine and Biology Society (pp. 6723-6726). Ieee.

o Rodrigo, M., Montesano, L., & Minguez, J. (2011, September). Classification of resting, anticipation and movement states in self-initiated arm movements for EEG brain computer interfaces. In 2011 Annual International Conference of the IEEE Engineering in Medicine and Biology Society (pp. 6285-6288). IEEE.

o Garipelli, G., Chavarriaga, R., & Millán, J. D. R. (2011, April). Single trial recognition of anticipatory slow cortical potentials: the role of spatio-spectral filtering. In 2011 5th International IEEE/EMBS Conference on Neural Engineering (pp. 408-411). IEEE.

o Gangadhar, G., Chavarriaga, R., & Millán, J. D. R. (2008). Fast recognition of anticipation-related potentials. IEEE Transactions on Biomedical Engineering, 56(4), 1257-1260.

o Garipelli, G., Chavarriaga, R., & del R Millán, J. (2013). Single trial analysis of slow cortical potentials: a study on anticipation related potentials. Journal of neural engineering, 10(3), 036014.

General comment 2: There are several portions of the experiment that need further details, so the experiment can be accurately reproduced by the readers; especially the takeover policy, a clear description of how the robot trajectories are defined (start, end, number of steps, etc), and how a trial is defined in the study. 

Response: The description of the experimental paradigm has been elaborated in section 2.2 Experimental Paradigm (lines 235-279). Takeover situations occur at the border between the two respective controlling areas in the sequential Collaboration (sC) scenario. If the trajectory spans two areas, then takeovers occur between each border crossing into the respective controlling area. Additionally, it is possible for the robot (or human) to make a mistake and return the robot end-effector to the opposing controlling area irrespective of the correct grid tile in the trajectory, at which point another takeover situation has occurred. In the intermittent Collaboration (iC) scenario, there are no clear areas defined for takeover situations, but rather the trajectory defines who should takeover controlling the robot. As a result, a takeover situation occurs when the robot or human is unable to move the robot end-effector to the next grid tile in the trajectory with the movement directions available. Thus, the partner must take over responsibility and continue the trajectory. In this scenario, it is possible for the robot (or human) to mistakenly takeover in a situation that does not complete the appropriate movement in the grid trajectory. At which point either the robot or human must correct the mistake depending on who has the responsibility based on the available movements. In this case, a takeover may be required to bring the robot back within the trajectory depending on the available movements and trajectory location. The description about target trajectories for the human and robot to follow have been elaborated in section 2.2 Experimental Paradigm. Start and end positions are randomly selected within the sub 5x5 grid. At least one direction change is required and 7 grid movements (trials) are required to complete the trajectory. The terms trial and episode are now defined upfront in section 2.2 Experimental Paradigm (line 239; line 245). 

General comment 3: The authors interpretation of the signal explored (ErrP vs. other possible signal) and the epoching/segmentation of the data leave gaps that need to be bridged. This affects all the paper sections, so it is recommended to revise them. It is especially important for the authors to explore the activity prior to the button push onset that represents a human takeover.

Response: In fact, the response to the event preceding the takeover situation was investigated. This should now be clarified with the extensive revisions we made to the introduction, description of experimental paradigm, and the interpretation of the observed signal in the discussion. We kindly refer the reviewer to our response to general comment 1 for further details.

General comment 4: Please proofread the whole document, some words are repeated, and minor corrections are needed. 

Response: Thank you for the pointer. We proof read the whole document and corrected any word repetitions or other forms of spelling/grammar mistakes. 

General comment 5: Finally, there seem to be some redundant references used throughout the document, such as over referencing prior work listing several publications of the authors for the same topic. 

Response: Thank you for the pointer. We have removed any redundant referencing of our prior work from the manuscript.

*** Abstract 

Specific comment 1: The authors mention in the abstract that this study allows “the prediction of the human anticipation of a robot-to-human takeover situation in single-trial.” but such outcome is yet to be completely accomplished. Specially since the classifier’s performance is nearly close to chance level. Moreover, it is important for the authors to clarify the real cognitive activity that is been study, which seems to be related to motor preparation, which indirectly reflects the future human takeover.

Response: We agree with the reviewer that, given the close-to-chance level accuracies, that this sentence may be misleading the reader. We decided therefore to delete it from the abstract. The interpretations about whether the observed effects result from cognitive versus motor related activity has been elaborated in the discussion (lines 691-727). 

*** Introduction

Specific comment 2: It is recommended for the authors to include in the “Neuroengineering approaches to HRC” section studies that have covered EEG measures of anticipatory behavior and motor preparation since these seems to be the core of the signal explored by the authors. 

Response: Thank you for this recommendation. The section “Neuroengineering approaches to HRI” now includes studies that have covered EEG measures of anticipatory behavior and motor preparation and puts them in relation to studies using ErrPs (lines 129-150). For further information we kindly refer the reviewer to our response to general comment 1. 

Specific comment 3: The authors must include/explain how the ErrP signal represents elements related to an anticipated takeover. Is this because a takeover represents an error and hence the ErrP will label such discrepancy? Please clarify. 

Response: We did not intent to convey that the ErrP signal represents anticipatory elements, but rather meant an ERP may contain such information. We have thoroughly revised the introduction and adjusted the terminology to remove any possibly confusing statements. 

Specific comment 4: [Page 2, line 22] Please make clear if footnote 1 is taken from reference 7 or if it is the authors’. It is not clear based on how reference 7 is used in the manuscript (not in the footnote). 

Response: The first author of reference 7 is one of the co-authors of the present manuscript (Emmanuel Dean Leon). To clarify the matter, we added reference 7 after the statement which was previously a footnote and is now moved to the main text (lines 22-27). 

Specific comment 5: [Page 4, line 121] The authors mention the definition for ErrP from reference 41, dating 1997. It is recommended to incorporate the seminal work of Falkenstein et al., 1991 and Gehring et al., 1993 (reference 44 in the manuscript); who first identified and described the ErrP signal, later known as “response ErrPs”. Reference 41 in this manuscript, Miltner et al., 1997, described what are called “feedback ErrP”. 

• Falkenstein, M. et al., 1991. Effects of crossmodal divided attention on late ERP components. II. Error processing in choice reaction tasks. Electroencep. Clin. Neurophysiol., 78, pp.447–455.

• Gehring, W.J. et al., 1993. A neural system for error detection and compensation. Psychol. Science, 4(6), pp.385–390.

Response: Thank you for the pointer. These references are now included in the definition of the ErrP (line 134).

*** Experimental HRC study

Specific comment 6: [Page 6, line 208] The authors seem to refer to the intermittent collaboration, so the figure referred seems to be Fig 1(b), not Fig 1(a). Please revise. 

Response: Thank you for the pointer. This has been corrected accordingly.

Specific comment 7: [Page 6. Line 216] The authors must include as part of the experimental paradigm an explanation of the criteria to select the positioning of the start and the end/target location, including the minimum number of steps required to reach the goal from the starting point (including metrics about the average and S.D. of the steps required), as well as how the spatial distribution of starting and ending points where balanced to avoid biasing the experiments. The authors mention this info is in reference 60 but such document is not currently available online (not Transactions on Biomedical Engineering, 2021). For reproducibility purposes, it is recommended to include this information in the manuscript (also please revise any typos in reference 60).

Response: The start, end and trajectory generation are now elaborated in section 2.2 Experimental paradigm. In short, the start and end positions were randomly chosen within a subset of the full grid (inner 5x5 grid) such that at least one direction change was present and 7 movements (trials) were required to complete the trajectory. An average of 7.19 +/- 0.39 grid tiles were selected for each trajectory. Start and end positions were selected at random from a uniform distribution within the 5x5 grid and thus did not bias any direction or location within the grid (lines 235-247). Reference 60 referred to unpublished work and is now removed from the manuscript.

Specific comment 8: [Page 7. Line 237] For reproducibility purposes, it is recommended for the authors to include the positioning of the head (facing forward and parallel to the horizontal plane?) and their visual angles when following the robot’s possible trajectories. 

Response: This recommendation has been added to section 2.3.1. Subjects were all of different height and thus the visual angle was not necessarily consistent between subjects. Additionally, while subjects were asked to relax and retain a comfortable posture, it is likely that subjects altered their head position throughout the experiment. As a result, based on the experiment setup dimensions, screen width and average female and male heights, a range of 30-35 degrees with respect to the monitor was determined to be the visual angle range for subjects and is now added to the manuscript (lines 288-289).

Specific comment 9: [Page 7. line 243] If motor preparatory activity is not the focus on the study, the author must explain what conditions were taken into account to avoid EEG motor activity, and preparatory activity, that could affect the EEG recorded during the task. 

Response: Thank you for this comment. We have thoroughly revised the introduction and discussion, specifically the interpretation of the observed effects in light with a possible overlap with motor preparatory activity. The revised line of argumentation does not exclude motor preparatory activity as part of the anticipatory EEG measures that we observe and utilize for robot learning. 

Specific comment 10: [Page 7, line 249] For reproducibility purposes, the authors must clearly explain what the rules (beyond the key distribution in the iC scenario) for takeover were (takeover policy), both for the subjects and the robot. In the sequential collaboration, can the robot alternate directions or will always follow one direction and only change once reaching some established goals? In the intermittent collaboration, will the robot or human have a maximum number of possible takeovers? Can takeovers happen one after the other or each agent can and must at least control two consecutive steps? Please clarify in the manuscript. 

Response: This has been clarified in section 2.2 Experimental Paradigm. In the intermittent Collaboration (iC) scenario, both the human and robot had the ability to takeover at any point throughout the task. However, each agent had control of only two of the four movements. Therefore, in cases where the human or robot was unable to complete the next grid position within the trajectory, a takeover needed to occur to continue the trajectory. Beyond this, the robot (or human) could mistakenly takeover in situations when it was not necessary for a takeover. Robot mistakes occurred at a rate of 30%. Not all mistakes resulted in a takeover from the human or even forced a subsequent takeover to occur. The resulting correction could still be the responsibility of the robot, but it could also be the responsibility of the human depending on the direction the robot moved with respect to the next grid position in the trajectory (lines 235-279).

Specific comment 11: [Page 8, line 290] It is recommended for the authors to clarify in section 2.3.4 “Timing validation“ and in the S1 Appendix (“Validation of robot motion timing” [Page 21, line 813]), if the validation experiment was performed per subject (specifying if done at the start or end of the recordings) or if it only took place once, with one subject, for the whole study across subjects. This clarification informs about the reliability of such validation. 

Response: The timing validation was completed in a single setting prior to any subject participation in the experiment. This clarification has been added to section 2.3.4 (lines 357-358). A single timing validation ensured no system latencies between parallel port and robot and ROS communication. Once validated, we expected no reason for any time shifts during subject experiments as the experiment and robot movement definitions were not altered.

Specific comment 12: [Page 8, line 293] The authors mention the concept of ‘anticipated’ but do not specifically explain what they mean until it is mentioned in the caption for Fig. 2. Please include in the ‘Experimental protocol’ a clear definition of what ‘anticipated’ means in the context of the experiment. 

Response: Thank you for this comment. We have thoroughly revised the description of the experiment and explicitly added a definition of the concept of “anticipated” at the end of section 2.2 (lines 266-279).

Specific comment 13: [Page 8, line 295] When the authors refer to “episodes”, it is recommended to properly explain what these represent. Are the episodes the possible directions the robot can take (4 in total) or do they represent a single trial (as later presented in table 1)? This is not fully clear. This information is necessary to replicate the methodology. 

Response: Thank you for the pointer. Episodes are now defined in section 2.2 Experimental Paradigm (line 245) in greater detail in section 2.4 Experimental Protocol (lines 363-366). To summarize, an episode represents the completion from start to end of a single trajectory. A trial, on the other hand, is a single robot movement from one grid space to the next. There are multiple trials within one episode of the experiment. A minimum of 7 trials is required per each episode (the shortest possible trajectory with no mistakes by either the human or robot). Once the episode is complete (the goal position is reached in the trajectory), a new episode begins with a new starting position, end position and trajectory between the two positions. The robot automatically moves to the starting position at which point the episode begins.

Specific comment 14: [Page 8, line 294] Please include in the manuscript information about the average time (and standard deviation) required to complete a trial, for each one of the scenarios. This information sheds light on the overlap of the epochs extracted, as well as on the total time experiment time (seems only available in page 16, line 580).

Response: Thank you for pointing this out. The trial duration was measured as the time difference between the onset of the robot's motion and its arrival within a 2mm radius of the next tile center. In section 2.3.1 we added that the average duration for the sequential Collaboration (sC) scenario is 875.88+/-133.91 ms (M+/-SD) and 872.34+/-131.35 ms (M+/-SD) for the intermittent Collaboration (iC). This is shorter than the 1200 ms time window and can cause overlapping if the robot starts moving immediately to the next tile. However, the central measure, the P300 ERP component, remains unaffected in all cases and never experiences overlapping. This is now added to section 2.3.1 Experimental setup (lines 310-312).

Specific comment 15: [Page 9, line 312, Section 2.5.1] It is necessary for the authors to include the length of the trials to fully understand its possible effects on the data epoching/segmentation. Under the consideration of a trial length too short, there is risk of high overlap between trial t and t+1 using the 1200ms epoch length. 

Response: Please see response to Specific comment 14.

Specific comment 16: [Page 9, line 336] The authors must clearly explain the evidence supporting an analysis of the before and after epochs is the appropriate approach for exploring the human and robot takeover. It is not clear why the authors do not evaluate the activity prior to the human or robot takeover instead. 

Response: Thank you for the comment. In fact, we did evaluate the activity prior to the human or robot takeover, see Figure 2 and 3. The ERPs prior to the takeover/non-takeover situations were central to our analysis and only ERPs preceding a takeover/non-takeover situation were assessed for their predictive power. 

Specific comment 17: [Page 9, Table 1. and lines 313] In table 1 the authors us the variable (t) to represent the action before and after takeover or non-takeover, but later in section 2.5.1 (t) is used to represent the time in the epochs. This can be confusing for the reader. Please use different variables. 

Response: Thank you for the pointer. In line with our response to general comment 1, we changed the terminology throughout the paper to tr-1 to indicate before and trial tr to indicate the trial after the takeover or non-takeover situation.

Specific comment 18: [Page 10, line 374] In section 2.5.4 it would be useful, for reproducibility purposes, to make clear to the reader what the classification outcomes are, including what the two classes represent, the y (x) values {-1, 1} as well as subscripts 1 and 2 in the equations (for the means and covariances). 

Responses: Thank you for this comment. We hope to have understood the reviewer’s comment correctly. The indices 1 and 2 indicate class 1 and 2 which are placeholders for the specific classes explained below in subsection “Validation”. We have added an explanation to section 2.5.4 (lines 476-479) which explains upfront what the placeholders stand for and what a classifier decision with y(x) = -1 and y(x) = +1 represent. 

Specific comment 19: [Page 10, line 390] The authors are encouraged to make clearer what the classes of the different classification problems are. It is mentioned later in the ‘Validation’ paragraph, but this information should be explicit in the ‘Binary classification’ section.

Response: Thank you for the suggestion. We added explanations (lines 476-479) that provides the reader with clarification prior to the ‘Validation’ paragraph. See also response to specific comment 18.

Specific comment 20: [Page 11, line 408, Validation] Could the authors explain what effect has in the decoding performance the use of the pick with replacement approach rather than balancing the conditions through pick without replacement (acknowledging this later approach is not appropriate since it would lead to folds with less trials than those needed to train a decoder using a features vector of 27 elements)?

Response: To address the class-misbalance issue that can lead to biased classifiers, we used a standard minority class resampling approach. The authors do not see clearly the reasoning behind a pick without replacement approach, since with the given dataset this would rapidly lead to no more samples being available. To address the situation that some samples are – by chance – used more often relative to other samples, we repeated the 10-fold CV for various times and obtain an average measure. This ensures that all samples are used for approximately the same amount of times.

***Results of experimental HRC study

Specific comment 21: [Results] Please report the percentage of correct classification using false/true positives and false/true negatives, per subject, to better understand the effect of biased training data in the decoder performance. Moreover, please include in the results the overall classification chance per subject. 

Response: Thank you for this comment. In fact, we reported about true positive and true negative rates as per class ACC (e.g. ACC_HH, ACC_HR, ACC_RR, ACC_RH). The terminology might have been misleading and we therefore decided to change them to TNR_HH, TPR_HR, TNR_RR, TPR_RH throughout the manuscript. Per subject classification results are now detailed in the appendix S3, including the per subject specific sample-size adjusted chance-levels.

Specific comment 22: [Page 11, line 432, section 2.6.1] Are the analysis presented an aggregate of all available subjects or were they performed per subject? The authors must be clear about their approach and include in the manuscript if there were differences or similarities across subjects. This information is key for finding overall trends in the results presented. 

Response: The results, e.g. the plots represent the grand average across all 11 subjects. Similarities and dissimilarities across subjects are represented in the across subjects statistical test results described in detail in Appendix S2. Appendix S3 now contains individual ERP traces and topographic plot for each subject.

Specific comment 23: [Page 12, below line 469, Figure 2 caption] The authors must make clear if the data presented belongs to one subject or if it is an aggregate of all subjects. Overall, it is expected that the results will be those of a representative subject. 

Response: The presented results are the average across all 11 subjects (M +/- SD). Subject individual results are detailed in Appendix S3. To disambiguate the text, we added “average across subjects” (line 562) and replaced “Grand average” with “Average across subjects” in the caption of Table 2. 

Specific comment 24: [Page 12, below line 469, Figure 2 caption] The authors state in the caption that ‘anticipated’ in the experiment’s context refers to “the belief that a takeover situation will occur”. What is the evidence to make this statement? 

Please consider the following: In the sequential collaboration scenario, takeover by the human only occurs once, in the transition boundary. But takeover seems to be defined by specific rules (the takeover policy that needs to be specified in the manuscript). For the belief of a takeover in this scenario, such belief does not really exists given the boundary itself and the current trajectory defined by the robot are clear elements that determine if a takeover should, and can, take place or not. In other words, a takeover is a function of the X-Y coordinates of the robot, and its past trajectory. Uncertainty of action only exists in the intermittent collaboration scenario. But it is again guided by the takeover policy provided to the subjects and robots. Without full access to the rules established it is impossible to understand how such expectation could be the driver of the activity studied in this experiment. 

Response: Thank you for pointing this out. We address this comment by (1) elaborating the descriptions of the experimental setup, see Section 2.2. Experimental paradigm, and (2) by changing a few statements to resolve any remaining ambiguity, specifically in section 2.2 (lines 254-256) “… in which the responsibility changes at the border of the colored areas. This concrete definition of the border areas facilitates a mental preparation of the user to execute the takeover action.”, and in section 2.2 (lines 275-276) “The explicit definition of tasks and the time/place where they will be triggered fosters the mental preparation of the user and induces expectations and thus anticipations of upcoming takeover situations.”, as well as in the caption of Figure 2 “… and after (tr) an anticipated takeover situation. Anticipated in this context refers to the user’s mental preparation to the taking over action produced by the specific task allocation in the experimental protocol.”

Specific comment 25: [Page 13, figure 3 caption] The authors are recommended to include in the figure 3 caption if the data presented belongs to before (t) or after (t + 1) an anticipated takeover situation.

Response: Thank you for this recommendation. The data belong to before (tr-1) an anticipated takeover situation. This information is now more explicitly mentioned in caption of Figure 3.

Specific comment 26: [Page 13, line 474] The authors are encouraged to make clearer what the classes of the different classification problems are. If the two possible outcomes are HH or HR and RR or RH, please simplify the language and state this. Moreover, the style of presenting accuracy is confusing. The author present classification accuracy for one class and later for the other as if they were independent results. If these are related to true positive results for each possible class, please indicate it so but avoid using the ACCHH format since it misleads the reader to believe six different binary classifications took place. 

Response: Thank you for the pointer. We agree that the choice of ACC for per-class accuracies were misleading. We have changed the terminology throughout the manuscript to TNR and TPR for per class classification rate and spared ACC for the overall classification accuracy for each of the two binary classification problems. This is changed both in various location of the main text and more prominently in Table 2, Figure 4, and the tables in Appendix S3. 

Specific comment 27: [Page 13, Table 2] The authors are encouraged to change the format of the table, the names used for the column titles, and how the classification performance is presented. It is recommended to include the overall classification accuracy followed by the false and true positives, as well as false and true negatives. This allows better understanding of the classification and its tradeoffs. Please apply these recommendations to the S3 appendix information too.

Response: Thank you for the recommendation. We have made the recommended changes, see more detailed response to specific comment 26. 

Specific comment 28: [Page 13, line 482] To fully understand how close the simulation is to a real scenario, the authors are encouraged to clarify what scenario is simulated in this section, the sequential or intermittent collaboration. Please specify. 

Response: Thank you for the pointer. We simulated the intermittent collaboration, since the empirical decoding performance of this interaction type was used in the simulation. This is now specified in the description of the robot RL model (line 568-569) and in the description of SIM1 (line 622-623).

Specific comment 29: [Page 16, line 589] The authors used a theoretical value for the chance level (50%) while referring to using other empirical decoder parameters for their analysis. It is recommended to use the empirical chance level as well (54.3%) to truly “provide the closest estimate of how the robot reinforcement learning approach would perform in a real HRC experiment”. Upon making this change, it is recommended to update the figures affected by this value. 

Response: Thank you for this comment. The empirical (e.g. sample-size adjusted) chance level represents the 95% probability threshold, e.g. the boundary at which we can assume a 5% error when stating that an empirical value larger than this boundary is indeed larger than chance. The empirical chance level reported alongside the decoding performance results is only relevant in the context of the classification performance and the given number of samples used to compute these. Therefore, based on these explanations, we hope the reviewer agrees with us that there is no rationale in using the empirical chance level for the simulation. Instead, using 50% allows to simulate a truly random decoder. Since any value larger than 50% will eventually lead to some learning if allowing for enough time, it was important to proof – based on the empirical chance level – that the empirical decoding performance not just represents a random fluctuation from theoretical chance-level and thereby in fact result from a random decoder, but is indeed significant above chance-level. Because only a such a non-random decoded guarantees learning in the long run (as can be seen from the results of SIM2). 

Specific comment 30: [Page 16, Figure 4 caption] For panel (c) the text reads “…dynamic task re-assignment after every 300 trials…” but the figure shows re-assignment after 40 episodes, that seem to represent 400 trials instead (based on text in a “trial = episode*10”). Please clarify and correct if necessary. 

Response: Thank you for the pointer. It was indeed wrongly indicated in the figure caption and is now corrected accordingly. 

Specific comment 31: [Page 17, line 614] The authors state in the discussion section that “the neural responses observed and documented in the form of ERPs and ERSPs were surprisingly clean, e.g. comparable to screen-based stimuli which are easier to control” but it is not clearly defined if such signals represent grand averages of one specific subject or the overall average from 11 subjects, which radically modifies how clean the signals will look upon replication of the study. The authors are encouraged to clarify if the results are subject-specific and similar across subjects. 

Response: Thank you for this comment. According to our experience of signal quality and signal-to-noise ratio in screen-based studies, we do consider the signals observed in this study as surprisingly clean - surprising in the context of having been elicited by a real robotic manipulator which is more challenging to control in a time-precise manner than screen-based stimuli. Since the notion seemed to be misleading and did not add value to the interpretation of our signals, we deleted it from the discussion. Furthermore, we clarified in Figures 2 and 3 that plots represent the grand average across all 11 subjects and added Supplementary Figure 1 to Appendix S3, which shows the average ERP traces and topographic patterns of each individual subject. 

Specific comment 32: [Page 17, line 623] The authors state that “the ERP elicited … contains information about whether the human partner anticipated role changes of responsibility…” and base the before (t) and after (t+1) trial definition, and the subsequent results using this trial definition, to support their statement. Nevertheless, it is still not clear that the data analyzed indeed reflects an ERP that clearly contains anticipation. There is a wealth of research in EEG measures of anticipatory behavior and movement preparation that were not covered by the authors and that could clearly specify how to epoch the data to truly find ERPs and other signal characteristics to inform about the human partner intentions to takeover. The authors are encouraged to explore the EEG measures of anticipatory behavior and movement preparation literature. 

Response: Thank you for this comment. In line with our response to general comment 1, we have thoroughly revised the interpretation of our findings in the discussion (lines 691-727).

***Discussion

Specific comment 33: [Page 18, line 674] The authors state that “…in the present study, we investigate a predictive measure, e.g., a neuronal response informative of an anticipated situation, not the response to a situation that is currently happening. To the best of our knowledge, such kind of information retrieval from the human EEG for augmentation of HRC has rarely - if never - been proposed/investigated before.” It is not clear the authors have provided enough evidence that the signal studied is a “neural response informative of an anticipated situation” nor have they shown that such signal is not motor preparatory activity related to the following button press action. In this context, there is strong evidence that a motor preparatory signal exists and has been explored in the context of human-robot collaboration. Below the authors can find a set of articles covering this area of study. 

• Bozinovski and L. Bozinovska, "Anticipatory brain potentials in a Brain-Robot Interface paradigm," 2009 4th International IEEE/EMBS Conference on Neural Engineering, 2009, pp. 451-454, doi: 10.1109/NER.2009.5109330.

• R. Chavarriaga, X. Perrin, R. Siegwart and J. del R. Millán, "Anticipation- and error-related EEG signals during realistic human-machine interaction: A study on visual and tactile feedback," 2012 Annual International Conference of the IEEE Engineering in Medicine and Biology Society, 2012, pp. 6723-6726, doi: 10.1109/EMBC.2012.6347537.

• S. L. Norman, M. Dennison, E. Wolbrecht, S. C. Cramer, R. Srinivasan and D. J. Reinkensmeyer, "Movement Anticipation and EEG: Implications for BCI-Contingent Robot Therapy," in IEEE Transactions on Neural Systems and Rehabilitation Engineering, vol. 24, no. 8, pp. 911-919, Aug. 2016, doi: 10.1109/TNSRE.2016.2528167.

• Smyk NJ, Weiss SM and Marshall PJ (2018) Sensorimotor Oscillations During a Reciprocal Touch Paradigm With a Human or Robot Partner. Front. Psychol. 9:2280. doi: 10.3389/fpsyg.2018.02280

• Di Liberto, G.M., Barsotti, M., Vecchiato, G. et al. Robust anticipation of continuous steering actions from electroencephalographic data during simulated driving. Sci Rep 11, 23383 (2021). https://doi.org/10.1038/s41598-021-02750-w

Response: Thank you for the comment and suggestions. In line with general comment 1, we have thoroughly revised the introduction and discussion as well as the description of the experiment. Given the wealth of motor related anticipatory brain responses, missing from the initial manuscript and now included, we also removed any notion of our work being the first to study anticipatory brain responses in the context of BCI and HRI. All of above suggested references are now referenced in the introduction and the discussion of the manuscript.

Specific comment 34: [Page 19, line 728] The authors express that “…this neuronal measure provides information about the hidden states of a user, e.g., preferences and intentions. This novel information …”. It is important to clarify that the brain activity explored in this manuscript is likely the EEG desynchronization reflecting a motor preparation prior to a button push, which has been extensively study (see previous comment and the papers mentioned). For this reason, it is recommended for the authors to be careful using the concepts “preferences and intentions”, as well as “novel information”. 

Response: Thank you for the comment. In line with our response to general comment 1, we have thoroughly revised the discussion and conclusion and removed or adjusted any of the statements listed above and made clear that motor preparatory processes cannot be excluded as being part of or strongly linked with our observed anticipatory EEG measures, see discussion (lines 675-727) and conclusion (lines 981-996).

Specific comment 35: [Page 19, line 731] The comments expressed in the section “A neural correlate of anticipation of human-robot takeover situations?”, specially the first paragraphs, should be reassessed under the lens of the literature in motor anticipatory activity previously suggested. 

Response: This subsection of the discussion has been thoroughly revised and does now properly relate our work to the literature on motor preparatory activity, see first subsection of the discussion (lines 691-727). 

***Conclusion

Specific comment 36: The authors do explore neural correlates although its interpretation and the way the data is studied leaves gaps that need to be bridged. 

Response: Thank you for the pointer. We have thoroughly revised the interpretation of our findings in light with the literature on motor preparatory activity (lines 691-727).

Specific comment 37: [Page 20, line 790] The authors are invited to explore the use of motor preparation as a means for expressing an anticipatory takeover but paying careful attention to how the data is segmented and analyzed. 

Response: Thank you for the pointer. We have thoroughly revised the interpretation of our findings and clarified that residual motor preparatory activity cannot be excluded as part of our observed signal (lines 691-727). 

Specific comment 38: [Page 20, line 790] The authors are recommended to avoid using the statement “novel neuro-cognitive method” since the analysis of motor preparation, and their classification, in human-robot collaboration tasks has been previously explored. 

Response: We agree with the reviewer and have correspondingly removed this statement both from the conclusion (lines 981-996) and the abstract (page 1). 

***References

Specific comment 39: [Page 25] Check the order of the authors in the reference section. As an example, in reference 35 the first author is DelPreto, not Hasani. 

Response: Thank you for the pointer. The respective reference was corrected accordingly and all other references were double checked. 

Specific comment 40: Some references, especially those from 2021, were not available online. Please double check the full references and add (in press) if those references are not yet published. 

Response: We double checked all references for availability and removed former reference 60 which referred to unpublished work. 

Specific comment 42: [Page 27] Reference 60 was not found in the IEEE Xplore site nor online. Double check the title, authors, and journal.

Response: Apologies if this reference caused confusion. Reference 60 referred to unpublished work and should not have been referred to in the paper. It has been removed in the meantime. 

***Figures 

Specific comment 43: [Figure 2] The authors are encouraged to explore the EEG features of preparatory movement activity and mu-rhythms, which are very similar to what is presented in the bottom topoplot next to channel C4 in (b) 

Response: The findings on preparatory movement related activity and mu-rhythms presented in Figure 2, are now discussed in greater detail in the first subsection of the discussion (lines 675-727). 

Specific comment 44: [Figures 2 and 3] The authors are encouraged to clarify if the results presented belong to a single subject (if so specifying its number) or are an aggregate of data from all subjects. Moreover, for readability purposes, the text in the figure should be sufficient to understand the image. Hence, the authors need to include in the C4, C3, FC1, and T7 plots the title (ERSPs) as well as axis labels and values (seems it is frequency and time as in ref. 69 but this must be confirmed by the authors and made clear to the readers). Moreover, it is not clear what the meaning of the curly brackets is. It seems to be pointing up to a specific time in the grand average ERP trace above it. It can be confused as zooming into the data at that specific time (200ms in (b), 650ms in (c), 850ms in (d)). It is fundamental to include color bars that indicate the meaning of the different colors in the topoplots and the ERSPs plots. 

Finally, it is recommended for the authors to use the naming standard established in table 1. If ‘Robot takeover from human’ is represented by ‘HR’, then please include this ‘HR’ in the text on the left of the plot. The same applies for the ‘Human takeover from robot’ text at the bottom of the figure.

Response: Thank you for these detailed recommendations. We followed all of them for figure 2 and 3, specifically, we added to the figure caption that the content represents the aggregate, e.g. means of the data of all subjects; we added titles, axes labels and values to the ERSP plots; we removed the curly brackets to avoid confusion and added colorbars to the topographic plots and the ERSPs. The horizontally aligned headings also contain the abbreviations ‘HR’ and ‘RH’ in bold fonts now. 

Specific comment 45: [Figure 3] Please follow for figure 3 the same recommendations about axis labels, color bars, title, the curly brackets and the use of the naming standard ‘RH’ that was mentioned for Figure 2.

Response: Same as for Figure 2. See response to specific comment 44. 

Specific comment 46: [Figure 4] The author must clarify what an episode means and what it represents in the study, so it makes sense in the different Figure 4 plots. Moreover, the empirical values for ACC shown in (a) do not match the ones presented in table 2. Please clarify where these values come from and modify the plots as necessary using the correct values (e.g. based on Table 2, ACCHR is 51.4 for both iC or sC, respectively, not 56.1), including the empirical chance accuracy (54.7). 

Response: Thank you for the pointers. A clear definition of an episode is now provided in the detailed experiment description, specifically in section 2.2 Experimental Paradigm (lines 243-245). The empirical values for ACC were indeed wrongly indicated in the figure, but the underlying plot was computed based on the correct values reported in Table 2. The correct values are now indicated in the figure. The terminology of the empirical classification performance measures was also changed to TNR and TPR in accordance to specific comment 21. 

Specific comment 47: Extra figures: The authors are encouraged to present the topoplots, ERSPs and traces for the different subjects. 

Response: Thank you for the suggestion which we followed and added a supplementary figure to Appendix S3 depicting the ERP traces and topoplots of each individual subject. The distribution of individual time-domain and spatial patterns show highest consistency across subjects for ERP latencies around 500 ms. Figure 2 and 3 each contain a pointer to this new supplementary material to direct the interested reader. 

Reviewer 2:

I enjoyed this paper and I think it will be an important contribution. I would recommend to accept with minor revisions.

Comment 1: I read the task description and reviewed the figures on the task. It’s still unclear to me exactly what the task is and how collaboration and hand-offs happened between the human and the robot. I would suggest to add a lot more detail about this so future researchers can accurately reproduce this work or approximate tasks like it. 

Response: Thank you for this comment. To enable future researchers to reproduce this work we added further explanations of the task and the episodes in section 2.2 Experimental Paradigm (lines 235-279) and section 2.4 Experimental Protocol (lines 363-366). In brief, we explained in more detail that the task was to follow the trajectory from the start to the end. An episode represents the completion from start to end of a single trajectory. A trial, on the other hand, is a single robot movement from one grid space to the next. There are multiple trials within one episode of the experiment. A minimum of 7 trials is required per each episode (the shortest possible trajectory with no mistakes by either the human or robot). Once the episode is complete (the goal position is reached in the trajectory), a new episode begins with a new starting position, end position and trajectory between the two positions. The robot automatically moves to the starting position at which point the episode begins. In the sequential Collaboration, hand-offs (i.e. takeover) happened at the border. In the intermittent Collaboration, takeover occurred depending on the direction of the next trial and the assigned responsibility to the human and the robot.

Comment 2: The paper emphasizes it is unique because it uses anticipatory human EEG signals to guide human-robot collaboration. Previous research (see below) has focussed on using the occurrence of errors to guide behavior and how this impacts construct like human-robot and human-automation trust. In that research, the error signals associated with the discrepancy between expected and actual behavior is what is most predictive and could be used to guide machine behavior in a neuroadaptive system. However, given the results presented in this paper, this raises the intriguing possibility that several signals (anticipatory and reactive) could be combined to improve HRI. For example, earlier fMRI work suggested that an “intention to trust” signal moved from a reactionary signal to an anticipatory one (King-Casas et al., 2005). Using the literature below, I would suggest to add a point (the relationship between anticipatory and reactive signals) like this in the discussion.

• [neuroadaptive systems] Zander, T. O., Krol, L. R., Birbaumer, N. P., & Gramann, K. (2016). Neuroadaptive technology enables implicit cursor control based on medial prefrontal cortex activity. Proceedings of the National Academy of Sciences, 113(52), 14898-14903 

• [trust] King-Casas, B., Tomlin, D., Anen, C., Camerer, C. F., Quartz, S. R., & Montague, P. R. (2005). Getting to know you: reputation and trust in a two-person economic exchange. Science, 308(5718), 78-83 

• [monitoring automation] - Berberian, B., Somon, B., Sahaï, A., & Gouraud, J. (2017). The out-of-the-loop Brain: a neuroergonomic approach of the human automation interaction. Annual Reviews in Control, 44, 303-315

• [human vs automation] - Somon, B., Campagne, A., Delorme, A., & Berberian, B. (2019). Human or not human? Performance monitoring ERPs during human agent and machine supervision. Neuroimage, 186, 266-277 

• [detecting error] - Fedota, J. R., & Parasuraman, R. (2010). Neuroergonomics and human error. Theoretical Issues in Ergonomics Science, 11(5), 402-421 

• [monitoring automation / trust] - De Visser, E. J., Beatty, P. J., Estepp, J. R., Kohn, S., Abubshait, A., Fedota, J. R., & McDonald, C. G. (2018). Learning from the slips of others: Neural correlates of trust in automated agents. Frontiers in human

• [detecting error] Weller, L., Schwarz, K. A., Kunde, W., & Pfister, R. (2018). My mistake? Enhanced error processing for commanded compared to passively observed actions. Psychophysiology, 55(6), e13057 * 

• [trust] Akash, K., Hu, W. L., Jain, N., & Reid, T. (2018). A classification model for sensing human trust in machines using EEG and GSR. ACM Transactions on Interactive Intelligent Systems (TiiS), 8(4), 1-20 * 

• [trust] Wang, M., Hussein, A., Rojas, R. F., Shafi, K., & Abbass, H. A. (2018, November). EEG-based neural correlates of trust in human-autonomy interaction. In 2018 IEEE Symposium Series on Computational Intelligence (SSCI) (pp. 350-357). IEEE 

• [trust] Choo, S., & Nam, C. S. (2022). Detecting Human Trust Calibration in Automation: A Convolutional Neural Network Approach. IEEE Transactions on Human-Machine Systems 

• [trust] Goodyear, K., Parasuraman, R., Chernyak, S., de Visser, E., Madhavan, P., Deshpande, G., & Krueger, F. (2017). An fMRI and effective connectivity study investigating miss errors during advice utilization from human and machine agents. Social neuroscience, 12(5), 570-581 

• [trust] Goodyear, K., Parasuraman, R., Chernyak, S., Madhavan, P., Deshpande, G., & Krueger, F. (2016). Advice taking from humans and machines: An fMRI and effective connectivity study. Frontiers in Human Neuroscience, 10, 542

Response: Thank you for the comment and suggestions for additional literature. We added the following references to the section “Introduction - Open challenges in HRI”: King-Casas et al. 2005; de Visser et al. 2018; Akash et al. 2018; Choo & Nam 2022 (lines 32-96). Following the reviewer’s suggestion, we added a paragraph to the end of the discussion that covers the combination of various anticipatory, such as movement preparatory signals, and reactive/evaluative signals, such as ErrPs and other performance monitoring related signals, as well as measures of trust (lines 798-809). We have added several of above listed references to this section, specifically, de Visser et al. 2018, Akash et al. 2018, Somon et al. 2019, King-Casas et al. 2005, Goodyear et al. 2017, Wang et al. 2018, Choo & Nam 2022.

Comment 3: I would suggest to add the term “Human-Robot Interaction” to your work in the abstract or elsewhere in addition to the term HRC. The way you describe HRC is pretty much the description of the HRI field. HRC is more of a subcategory that describes how exactly human/robots/machines/automation should work together and coordinate, a subject of inquiry in both the HRI and human factors fields. This HRI term refers to the entire field / discipline (see HRI conference for example) and will make it so your article has broader appeal to the HRI community as well as the neuroscience communities and the neuroergonomic community. In addition some relevant literature from human factors / HRI: 

• [HRI and neuroscience] Henschel, A., Hortensius, R., & Cross, E. S. (2020). Social cognition in the age of human–robot interaction. Trends in Neurosciences, 43(6), 373-384 

• [transparency] Chen, J. Y., Lakhmani, S. G., Stowers, K., Selkowitz, A. R., Wright, J. L., & Barnes, M. (2018). Situation awareness-based agent transparency and human-autonomy teaming effectiveness. Theoretical issues in ergonomics science, 19(3), 259-282 

• [transparency] Mercado, J. E., Rupp, M. A., Chen, J. Y., Barnes, M. J., Barber, D., & Procci, K. (2016). Intelligent agent transparency in human–agent teaming for Multi-UxV management. Human factors, 58(3), 401-415 

• [Function allocation] De Winter, J. C., & Dodou, D. (2014). Why the Fitts list has persisted throughout the history of function allocation. Cognition, Technology & Work, 16(1), 1-11 

• [Function allocation] Kaber, D. B. (2018). Issues in human–automation interaction modeling: Presumptive aspects of frameworks of types and levels of automation. Journal of Cognitive Engineering and Decision Making, 12(1), 7-24 

• [adaptive automation] Parasuraman, R., Bahri, T., Deaton, J. E., Morrison, J. G., & Barnes, M. (1992). Theory and design of adaptive automation in aviation systems. Catholic Univ of America Washington DC cognitive science lab 

• [adaptive automation] Scerbo, M. (2007). Adaptive automation. Neuroergonomics: The brain at work, 239252 

• [adaptive automation] Parasuraman, R., Mouloua, M., Molloy, R., & Hilburn, B. (1993, May). Adaptive function allocation reduces performance cost of static automation. In 7th international symposium on aviation psychology (pp. 37-42)

Response: Thank you for the comment and additional suggestions for literature. Following your suggestion, we have decided to remove the term human-robot collaboration (HRC) altogether from the manuscript and stick with the term human-robot interaction (HRI) as there is no need to constrain our work on a subform of HRI. In addition, we added several of above suggested papers to the introduction of the manuscript to elaborate the introductory section on “Open challenges in HRI”: Chen et al. 2018; Scerbo 2007; and Parasuraman et al. 1993 (lines 32-96).

Editor:

Comment 1: Please ensure that your manuscript meets PLOS ONE's style requirements, including those for file naming.

Response: Thank you for the pointer. We made necessary revision to the style and format of the manuscript to conform to PLOS ONE’s style requirements, specifically, (i) we changed the subfigure referencing from small characters in parenthesis to capital letter without brackets, e.g. Fig 1(a) to Fig 1A. (ii) we adjusted the naming and referencing for supporting material according to PLOS ONE’s style requirements. (iii) We removed all footnotes and included them in the main text.

Comment 2: Please provide additional details regarding participant consent. In the Methods section, please ensure that you have specified (1) whether consent was informed and (2) what type you obtained (for instance, written or verbal). If your study included minors, state whether you obtained consent from parents or guardians. If the need for consent was waived by the ethics committee, please include this information.

Response: Participants provided written informed consent which is now specified in the Methods section (line 229).

Comment 3: Please remove any funding-related text from the manuscript and let us know how you would like to update your Funding Statement. Currently, your Funding Statement reads as follows: “The author(s) received no specific funding for this work.” Please include your amended statements within your cover letter; we will change the online submission form on your behalf.

Response: The statement “This work was partially supported by the Elite Network Bavaria (ENB) through the master program in neuroengineering (MSNE)” has been deleted from the manuscript to avoid any confound with the funding statement given upon initial submission. 

Comment 4: In your Data Availability statement, you have not specified where the minimal data set underlying the results described in your manuscript can be found. PLOS defines a study's minimal data set as the underlying data used to reach the conclusions drawn in the manuscript and any additional data required to replicate the reported study findings in their entirety. All PLOS journals require that the minimal data set be made fully available. For more information about our data policy, please see http://journals.plos.org/plosone/s/data-availability. Upon re-submitting your revised manuscript, please upload your study’s minimal underlying data set as either Supporting Information files or to a stable, public repository and include the relevant URLs, DOIs, or accession numbers within your revised cover letter. For a list of acceptable repositories, please see http://journals.plos.org/plosone/s/data-availability#locrecommended-repositories. Any potentially identifying patient information must be fully anonymized. Important: If there are ethical or legal restrictions to sharing your data publicly, please explain these restrictions in detail. Please see our guidelines for more information on what we consider unacceptable restrictions to publicly sharing data: http://journals.plos.org/plosone/s/data-availability#loc-unacceptable-data-access-restrictions. Note that it is not acceptable for the authors to be the sole named individuals responsible for ensuring data access. We will update your Data Availability statement to reflect the information you provide in your cover letter.

Response: The minimal data set consisting of the anonymized EEG and meta data has been uploaded to the following public repository: https://github.com/stefan-ehrlich/HRC_neurobased_taskplanning. This information is also updated in the manuscript (lines 1002-1004)

Comment 5: We note that you have indicated that data from this study are available upon request. PLOS only allows data to be available upon request if there are legal or ethical restrictions on sharing data publicly. For more information on unacceptable data access restrictions, please see http://journals.plos.org/plosone/s/data-availability#loc-unacceptable-data-access-restrictions. In your revised cover letter, please address the following prompts: a) If there are ethical or legal restrictions on sharing a de-identified data set, please explain them in detail (e.g., data contain potentially sensitive information, data are owned by a third-party organization, etc.) and who has imposed them (e.g., an ethics committee). Please also provide contact information for a data access committee, ethics committee, or other institutional body to which data requests may be sent. b) If there are no restrictions, please upload the minimal anonymized data set necessary to replicate your study findings as either Supporting Information files or to a stable, public repository and provide us with the relevant URLs, DOIs, or accession numbers. For a list of acceptable repositories, please see http://journals.plos.org/plosone/s/data-availability#locrecommended-repositories. We will update your Data Availability statement on your behalf to reflect the information you provide.

Response: See response to editorial comment 4. The minimal data set consisting of the anonymized EEG and meta data has been uploaded to the following public repository: https://github.com/stefan-ehrlich/HRC_neurobased_taskplanning. This information is also updated in the manuscript (lines 1002-1004)

Comment 6: Your ethics statement should only appear in the Methods section of your manuscript. If your ethics statement is written in any section besides the Methods, please delete it from any other section. 

Response: We deleted the ethics statement from the manuscript and added above requested information about type of consent to the Methods section (line 229).

Comment 7: Please review your reference list to ensure that it is complete and correct. If you have cited papers that have been retracted, please include the rationale for doing so in the manuscript text, or remove these references and replace them with relevant current references. Any changes to the reference list should be mentioned in the rebuttal letter that accompanies your revised manuscript. If you need to cite a retracted article, indicate the article’s retracted status in the References list and also include a citation and full reference for the retraction notice.

Response: We double checked all references and removed a reference ([60] in the initial manuscript) that referred to unpublished work. All changes to the list of references (mostly additions) are noted in the responses to the reviewer’s comments.

---

## [Decision Letter · Decision Letter 1]

28 Feb 2023

PONE-D-21-32176R1Neuro-cognitive measures of anticipated takeover situations enable human-robot collaborative task planningPLOS ONE

Dear Dr. Ehrlich,

Thank you for submitting your manuscript to PLOS ONE. After careful consideration, we feel that it has merit but does not fully meet PLOS ONE’s publication criteria as it currently stands. Therefore, we invite you to submit a revised version of the manuscript that addresses the points raised during the review process.

The academic editor would like to thank the authors' effort in improving the working paper during the review process. Yet there are issues raised by the reviewers that remain to be addressed. Please refer to the reviewers' comments and revise the manuscript accordingly.

We look forward to receiving your revised manuscript.

Kind regards,

Chun-Shu Wei

Academic Editor

PLOS ONE

Journal Requirements:

Additional Editor Comments (if provided):

The academic editor would like to thank the authors' effort in improving the working paper during the review process. Yet there are issues raised by the reviewers that remain to be addressed. Please refer to the reviewers' comments and revise the manuscript accordingly.

Reviewers' comments:

Reviewer's Responses to Questions

**Comments to the Author**

1. If the authors have adequately addressed your comments raised in a previous round of review and you feel that this manuscript is now acceptable for publication, you may indicate that here to bypass the “Comments to the Author” section, enter your conflict of interest statement in the “Confidential to Editor” section, and submit your "Accept" recommendation.

Reviewer #1: All comments have been addressed

Reviewer #3: All comments have been addressed

2. Is the manuscript technically sound, and do the data support the conclusions?

Reviewer #1: Yes

Reviewer #3: Partly

3. Has the statistical analysis been performed appropriately and rigorously? 

Reviewer #1: Yes

Reviewer #3: Yes

4. Have the authors made all data underlying the findings in their manuscript fully available?

Reviewer #1: Yes

Reviewer #3: Yes

5. Is the manuscript presented in an intelligible fashion and written in standard English?

Reviewer #1: Yes

Reviewer #3: Yes

6. Review Comments to the Author

Reviewer #1: All comments and suggestions have been addressed.

.........................................................................

Reviewer #3: Comments to the authors:

The authors investigate the characteristics of anticipatory movement-related brain activities in a human-robot interaction (HRI) scenario and how the neuro-cogitive measurements can improve robots' learning abilities. The manuscript is well written and organized. The authors have also addressed most of the comments from the previous reviewers. However, some of the details of experiment paradigm and definitions are still missing and thus reduce the value of the manuscript. I have some remarks for the authors.

1. I strongly recommend the authors to include a figure of experiment paradigm to better explain the experiment. When should the human/ robot initiate actions (key pressing)? Is there a signal telling human when one can initiate actions? Does robot always initiate actions at certain time point? If yes, is it right after the robot arm reaches the center of the grid or there is a delay? If no, what is the range of time period?

2. In Page 6, line 220. The authors mention the robot (or human) could mistakenly takeover in situations and robot mistakes occur at a rate of 30%. How does robot make mistakes if they are programmed to initiate actions only when it is necessary? Does it mean that there is a process to control when the robot should initiate actions for the incoming trial? If yes, what is the criteria and when the robot will initiate actions (back to comment 1)?

3. Following comment 2, Did the authors take out those mistake trials from both human and robot to prevent mixing anticipatory movement-related brain signals with error-related brain signals in the following analysis?

4. Related to comment 1. In page 7, line 264. The authors mention they randomized the robot arm movement duration to prevent subject habituate to robot arm movement onset. Does the robot arm move to next grid right after receiving actions from either robot or human, or there is a constant/ random delay between receiving actions and movement onset?

5. In page 8, line 293. The authors explain how artifact comtaminated EEG channels are removed by kurtosis and interpolated from neighboring channels. However, the definition of threshold 2% is unclear. Also, the interpolations method need to be further explained (linear, spline, etc.).

6. In page 9, line 344. The authors state the number of the trials in sC and iC scenario are approximately equal. Does it mean that the trajetory to the target is not the shortest path and thus it can pass the bolder where takeovers happen several times in sC scenario? I do not remember anything in the manuscript related to how the trajetory is defined.

7. In page 12, line 468. Should the sentence be "a robot takeover situation from human"?

8. In page 13, line 504. The authors explain the simulation setup. However, there is no clear definition of which subtasks robot should take responsibility for (what is the correct subtasks for robot?). Are the correct subtasks defined as the subtasks not taken by human?

9. Related to comment 8. In page 18, line 710. The authors claim that imbalanced assignment of subtasks does not affect the learning. I am curious if the robot can learn faster (achieve higher Acc in early episode) when human chooses only 2 out of 10 comparing to 8 out of 10 since the number of "correct subtasks" for robot is less in the later condition. If there is no difference, the authors statement can be strengthen.

7. PLOS authors have the option to publish the peer review history of their article (what does this mean?). If published, this will include your full peer review and any attached files.

Reviewer #1: **Yes: **Andrés F. Salazar-Gómez

Reviewer #3: **Yes: **Chi-Yuan Chang

---

## [Author Response · Author response to Decision Letter 1]

30 Apr 2023

Response to Reviewer 3 comments:

The authors investigate the characteristics of anticipatory movement-related brain activities in a human-robot interaction (HRI) scenario and how the neuro-cogitive measurements can improve robots' learning abilities. The manuscript is well written and organized. The authors have also addressed most of the comments from the previous reviewers. However, some of the details of experiment paradigm and definitions are still missing and thus reduce the value of the manuscript. I have some remarks for the authors.

1. I strongly recommend the authors to include a figure of experiment paradigm to better explain the experiment. When should the human/ robot initiate actions (key pressing)? Is there a signal telling human when one can initiate actions? Does robot always initiate actions at certain time point? If yes, is it right after the robot arm reaches the center of the grid or there is a delay? If no, what is the range of time period?

Response: Thank you for the comment. We updated Fig. 1 to explain both scenarios, trial-by-trial in more detail. The figure illustrates in the sequential Collaboration (sC) scenario, that human and robot initiate the actions only in their area of responsibility, marked with different colors. In the intermittent Collaboration (iC), the human and robot handover is based on the direction of the next trajectory tile. When the robot initiates the actions, a new movement starts once the robot has arrived within a 2mm radius at the center of the tile. For more details, please see the answer to comment 4. These details are elaborated in Section 2.2 Experimental Paradigm in various locations (lines 186-235, changes are marked in blue in manuscript with tracked changes).

2. In Page 6, line 220. The authors mention the robot (or human) could mistakenly takeover in situations and robot mistakes occur at a rate of 30%. How does robot make mistakes if they are programmed to initiate actions only when it is necessary? Does it mean that there is a process to control when the robot should initiate actions for the incoming trial? If yes, what is the criteria and when the robot will initiate actions (back to comment 1)?

Response: The robot is programmed to initiate actions according to the answer to comment 1 (see above). In addition, the robot is programmed to randomly make mistakes (move in another direction than indicated by key press or “intended” by the robot) at a probability of 30%, independently if the next movement should have been initiated by the human or the robot. However, the brain responses to these mistakes are not the focus of the current study (please see the answer to the next question).

3. Following comment 2, Did the authors take out those mistake trials from both human and robot to prevent mixing anticipatory movement-related brain signals with error-related brain signals in the following analysis?

Response: That is correct; only non-error trials were using for the analyses performed in this work to avoid mixing anticipatory brain signals with error-related brain signals. This information is now added more prominently in Section 2.5.1, page 9, line 347.

4. Related to comment 1. In page 7, line 264. The authors mention they randomized the robot arm movement duration to prevent subject habituate to robot arm movement onset. Does the robot arm move to next grid right after receiving actions from either robot or human, or there is a constant/ random delay between receiving actions and movement onset?

Response: Thank you for addressing this point. We added additional clarity around robot movement duration (page 8, line 271). The robot movement duration was randomly set at the beginning of a session in the range of 0.6-1.2s. The movement duration is defined based on a start and end position at the center of grid spaces. However, additional random deviations from this movement duration arise from the robot completing a trial once within the 2mm radius of the end position (page 8, line 272), which depends on variations of preceding events. For example, a human may be slower to respond to the robot reaching the end position compared to the robot directly making movements. Once the robot is within the 2mm radius of the center of the grid, a trial is considered complete. At which point, a new trial begins with no delay (either robot takes action or human takes action after reacting). Therefore, the combination of preceding events and the 2mm radius make up the random robot movement duration per trial.

5. In page 8, line 293. The authors explain how artifact comtaminated EEG channels are removed by kurtosis and interpolated from neighboring channels. However, the definition of threshold 2% is unclear. Also, the interpolations method need to be further explained (linear, spline, etc.).

Response: Thank you for this comment. We clarified the statement by replacing the sentence with: “Next, we identified contaminated EEG channels using normalized kurtosis with a threshold of 2 (std. dev.) and subsequently used spherical interpolation to reconstruct rejected channels from the signals of neighboring electrodes.” (see page 8, line 302).

6. In page 9, line 344. The authors state the number of the trials in sC and iC scenario are approximately equal. Does it mean that the trajetory to the target is not the shortest path and thus it can pass the bolder where takeovers happen several times in sC scenario? I do not remember anything in the manuscript related to how the trajetory is defined.

Response: Thank you for the pointer. The trajectory to the target was not the shortest path, but it was randomly designed to contain at least one direction change (for example, first going to the right and afterwards going down) and at least 7 tiles. We expanded the parts in 2.2 Experimental paradigm to make it clearer (see pages 5-6, lines 194, 204, 221).

7. In page 12, line 468. Should the sentence be "a robot takeover situation from human"?

Response: Thank you for the pointer. That is correct and has been changed accordingly (see page 12, line 481)

8. In page 13, line 504. The authors explain the simulation setup. However, there is no clear definition of which subtasks robot should take responsibility for (what is the correct subtasks for robot?). Are the correct subtasks defined as the subtasks not taken by human?

Response: The human choses a random subset of the n available subtasks unbeknownst to the robot, whereby the number of subtasks per partner are predefined in all simulations. The robot assigned subtasks are naturally the remaining subtasks which the human has not chosen. The robot’s task is to infer the assigned subtasks from the human EEG anticipatory responses. We changed the following sentence to the paragraph to clarify the explanations: “R's assigned subtasks are naturally the remaining subtasks which the human has not chosen.” (see page 14, line 518).

9. Related to comment 8. In page 18, line 710. The authors claim that imbalanced assignment of subtasks does not affect the learning. I am curious if the robot can learn faster (achieve higher Acc in early episode) when human chooses only 2 out of 10 comparing to 8 out of 10 since the number of "correct subtasks" for robot is less in the later condition. If there is no difference, the authors statement can be strengthen.

Response: Thank you for this comment. In the example presented in the paper, the simulated human indeed chose 2 subtasks and left 8 to the robot (see line 635). We have run many variants of the misbalanced subtask assignment and generally observed “steeper robot learning curves when more subtasks are assigned to the robot compared to the human and vice-versa more shallow learning curves when less subtasks are assigned to the robot” (see line 631). As for the illustration of this result in the manuscript we found the example of 2 subtasks chosen by the human to be most illustrative which is why we decided to show it in the manuscript. To avoid any misunderstanding, we added a statement to the caption of figure 4 “(6 subtasks with balanced assignment, 10 subtasks with balanced assignment, and 10 subtasks with 2 human and 8 robot assigned subtasks)”.

---

## [Decision Letter · Decision Letter 2]

2 Jun 2023

PONE-D-21-32176R2Human-robot collaborative task planning using anticipatory brain responsesPLOS ONE

Dear Dr. Ehrlich,

Thank you for submitting your manuscript to PLOS ONE. After careful consideration, we feel that it has merit but does not fully meet PLOS ONE’s publication criteria as it currently stands. Therefore, we invite you to submit a revised version of the manuscript that addresses the points raised during the review process.

As the manuscript has undergone significant improvements, the authors are encouraged to address the minor issues mentioned in the reviewer's comments.

We look forward to receiving your revised manuscript.

Kind regards,

Chun-Shu Wei

Academic Editor

PLOS ONE

Journal Requirements:

Additional Editor Comments:

As the manuscript has been significantly improved, the authors are encouraged to address to the minor issues according to the reviewer's comment.

Reviewers' comments:

Reviewer's Responses to Questions

**Comments to the Author**

1. If the authors have adequately addressed your comments raised in a previous round of review and you feel that this manuscript is now acceptable for publication, you may indicate that here to bypass the “Comments to the Author” section, enter your conflict of interest statement in the “Confidential to Editor” section, and submit your "Accept" recommendation.

Reviewer #3: All comments have been addressed

2. Is the manuscript technically sound, and do the data support the conclusions?

Reviewer #3: Yes

3. Has the statistical analysis been performed appropriately and rigorously? 

Reviewer #3: Yes

4. Have the authors made all data underlying the findings in their manuscript fully available?

Reviewer #3: Yes

5. Is the manuscript presented in an intelligible fashion and written in standard English?

Reviewer #3: Yes

6. Review Comments to the Author

Reviewer #3: All my comments have been addressed properly. However, Figure 1 is missing in the revised manuscript.

7. PLOS authors have the option to publish the peer review history of their article (what does this mean?). If published, this will include your full peer review and any attached files.

Reviewer #3: **Yes: **Chi-Yuan Chang

---

## [Author Response · Author response to Decision Letter 2]

5 Jun 2023

Dear Editor and reviewers,

We thank the editor and reviewers for their constructive and insightful comments and recommendations which have helped further to significantly improve the manuscript. We have revised the manuscript (R3) according to the single left open comment from reviewer 3. Furthermore, we reviewed our reference list to ensure that it is complete and correct. Please note, that since the manuscript did not undergo any further changes, as the single reviewer 3 comment referred to Figure 1, the ‘Revised Manuscript with Track Changes’ is identical to the ‘Manuscript’.

Reviewer 3:

All my comments have been addressed properly. However, Figure 1 is missing in the revised manuscript.

Response: The figure was indeed missing from the submission – we missed to notice that it was not properly processed by the EM pdf maker. Apologies for rendering the need for another round of revision. The figure is now accompanying the revision and addresses all points raised by the reviewer in the previous revision.

---

## [Editor Report · Decision Letter 3]

19 Jun 2023

Human-robot collaborative task planning using anticipatory brain responses

PONE-D-21-32176R3

Dear Dr. Ehrlich,

We’re pleased to inform you that your manuscript has been judged scientifically suitable for publication and will be formally accepted for publication once it meets all outstanding technical requirements.

Kind regards,

Chun-Shu Wei

Academic Editor

PLOS ONE

Additional Editor Comments (optional):

The manuscript titled "Human-robot collaborative task planning using anticipatory brain responses" is recommended to be accepted for publication.
---

## [Editor Report · Acceptance letter]

2 Jul 2023

PONE-D-21-32176R3 

Human-robot collaborative task planning using anticipatory brain responses 

Dear Dr. Ehrlich:

I'm pleased to inform you that your manuscript has been deemed suitable for publication in PLOS ONE. Congratulations! Your manuscript is now with our production department. 

Kind regards, 

on behalf of

Dr. Chun-Shu Wei 

Academic Editor

PLOS ONE